# VARIATIONAL MODEL MERGING FOR PARETO FRONT ESTIMATION IN MULTITASK FINETUNING

## ABSTRACT

We propose a new variational model merging method that can yield arbitrarily accurate Pareto fronts in multitask finetuning. The idea is to first compute posterior-approximations on each task separately and then quickly merge them to obtain a cheap estimate of Pareto front. The main theoretical result is to show that more flexible posteriors necessarily yield better estimates, for example, a Pareto front obtained by merging full Gaussian posteriors is expected to be better than those obtained with isotropic Gaussians. This is because the error incurred by a specific class of distributions can always be reduced by increasing the size of the class. We validate the theory through extensive empirical results on deep networks (Vision and Language Transformers) where better Gaussian families consistently yields better or comparable Pareto fronts. Our work is a rare instance where Bayesian ideas are used to improve Pareto analysis.

## 1 INTRODUCTION

The Pareto front in multitask finetuning is important to give practitioners choices on trade-offs and to understand the relationships between tasks before deciding weightings that get the desired performance (Liu et al., 2023; Xu et al., 2024; Chung et al., 2024). Similarly to the traditional multitask learning (Caruana, 1997; Ruder, 2017), weighting is useful in tackling data imbalance, task interference, negative transfer, and also effects of variable task difficulty (Raffel et al., 2020; Liu et al., 2023). When left unresolved these can lead to issues, for instance, regarding safety (Jan et al., 2024). Weighting is especially relevant for mid-training which has recently been used to adapt LLMs during later training stages, for example, for multi-lingual transfer and to improve reasoning skills (Aghajanyan et al., 2021; Gemma Team, 2024a; Martins et al., 2024; Fujii et al., 2024).

However, obtaining the complete Pareto front is difficult, more so for large models where an exhaustive search over weights is out of question. Even if we could try a few weightings configurations, which ones should we try? There is no guarantee they would achieve the trade-off we want. Weights are often chosen arbitrarily and sometimes heuristically but these are not sufficient; see for example, Liu et al. (2023, Sec. 6). The weighting methods used for deep learning and pretraining can be adapted to search for a specific trade-off (Ren et al., 2018; Chen et al., 2018; Raffel et al., 2020; Groenendijk et al., 2021; Du et al., 2022; Yan et al., 2022; Xie et al., 2023; Thakkar et al., 2023), but a quick guidance on reasonable search areas or an approximate solution to start from would still be useful.

In this paper, to estimate the multitask finetuning Pareto front we leverage model merging to get quick and accurate approximations. Our main contribution is a new approach called variational model merging which can be used to design new merging strategies that yield better approximations that cover a wider range of the Pareto front by using better posterior approximations. This differs fundamentally from most previous works which only focus on the single best-performing weights, meaning a single model (Don-Yehiya et al., 2023; Jiang et al., 2023; Feng et al., 2024; Stoica et al., 2024; Yang et al., 2024). We show that such strategies do not always yield good-quality models, because they use less flexible posterior approximations; see Fig. 1b left, for an illustration. For instance, Task Arithmetic (Ilharco et al., 2023) corresponds to isotropic Gaussian posteriors, while better Hessian-based methods (Daheim et al., 2024) employ more flexible Gaussian forms.

More closely related to our work, Li et al. (2025a) also use model merging and quadratic surrogates of the evaluation metric. In contrast, our surrogates are automatically derived using variational learning and can be extended to more flexible approximations beyond quadratic ones. For example,

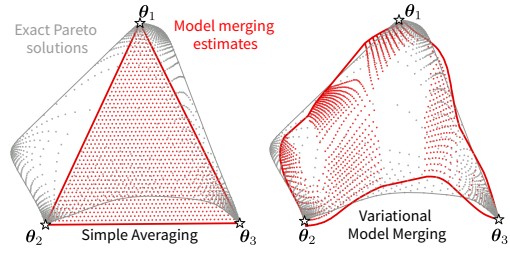

(a) Model merging to estimate the Pareto Set      (b) Variational Merging makes better estimates

Figure 1: An illustration of Variational Model Merging for estimating the Pareto set solutions of a toy multitask-learning problem with 3 tasks. The losses $\ell_t$ are defined over a 2-D parameter ($\boldsymbol{\theta}$) space and are weighted by $\alpha_t$ which is varied in a fixed grid over $[0, 1]$. The approximation on the left shows that simple averaging $\sum_t \alpha_t \boldsymbol{\theta}_t$ gives poor Pareto set estimates (red region) of the true Pareto set (grey contour). Each dot corresponds to a solution obtained by weighting tasks differently. Quality is improved with merging strategies that use more flexible posteriors, for example, mixture of Gaussians as shown on the right, albeit for a slight increase in cost.

we use variational model merging to introduce mixture-of-Gaussians merging as an example on how exponential-family posteriors can be leveraged to derive new merging recipes with better approximations. As shown by our results, this can lead to more accurate Pareto fronts by yielding models that are closer to the true Pareto solutions but with slightly higher costs (Fig. 1b right). We validate our methodology on several benchmarks on vision and language transformers. For example, we experiment with image classification using Vision Transformers (ViTs) (Dosovitskiy et al., 2021) of 86M parameters and adding new languages to GEMMA-2B (Gemma Team, 2024b) for machine translation. Our work exploits the power of Bayesian methods and model merging to quickly and accurately explore trade-offs models in the Pareto front of multitask finetuning.

## 2 WEIGHTED MULTITASK FINETUNING AND MULTI-OBJECTIVE LEARNING

Multitask finetuning aims to finetune a model $\boldsymbol{\theta}$ on several tasks $t = 1, \ldots, T$ at the same time. Denoting the loss of each task by $\ell_t(\boldsymbol{\theta})$ and the regularizer by $\mathcal{R}_0(\boldsymbol{\theta})$ we use a weighted loss:

$$\sum_{t=1}^{T} \alpha_t \ell_t(\boldsymbol{\theta}) + \mathcal{R}_0(\boldsymbol{\theta}), \text{ where } \alpha_t > 0 \text{ for } t = 1, 2, \ldots, T, \tag{1}$$

where we denote the loss-weight vector by $\boldsymbol{\alpha} = (\alpha_1, \alpha_2, \ldots, \alpha_T)$ and the finetuned parameters obtained with the weighting by $\boldsymbol{\theta}_{\boldsymbol{\alpha}}$. We will generally assume that $\alpha_t$ sum to 1 but this is not strictly required. Such multitask finetuning has recently become important for LLM alignment and usability and is widely used especially for later stages of pretraining, for example, to improve instruction-following abilities (Chung et al., 2024; Ouyang et al., 2022), for safety tuning (Gemma Team, 2024a), combining coding tasks (Liu et al., 2023), and mixing coding and math skills into LLMs, which is useful even when they are designed for other tasks like machine translation (Martins et al., 2024).

In practice, there are many reasons why choosing $\boldsymbol{\alpha}$ carefully is important (Caruana, 1997; Ruder, 2017). For instance, data might be imbalanced both in terms of information content and quality. There is also task interference, for example, a model that does math well, may not necessarily be the best at languages. Additionally, learning some tasks might hurt the performances on the other tasks, and then there is variability in task difficulty: some tasks are harder to learn and we do not want them to impact the tasks that are relatively easier to learn. The effects of these issues are often felt in practice. For example, adding too much safety data can make the model too conservative (Bianchi et al., 2024); too much instruction finetuning can undo safety alignment and open new vulnerabilities (Qi et al., 2024; Jan et al., 2024). Such problems can be avoided by careful task weighting.

Multi-objective learning (Sener & Koltun, 2018) could be used to guide task weighting. Intuitively, we would like to find weights $\boldsymbol{\theta}$ of models that are not worse on every task than another model $\boldsymbol{\theta}'$.

This concept is formalized by Pareto dominance, where:

$$\boldsymbol{\theta} \succ \boldsymbol{\theta}' \text{ iff } \ell_{t'}(\boldsymbol{\theta}) < \ell_{t'}(\boldsymbol{\theta}') \text{ for at least one } t' \text{ and } \ell_t(\boldsymbol{\theta}) \leq \ell_t(\boldsymbol{\theta}') \text{ for all } t = \{1, 2, \ldots, T\}. \qquad (2)$$

indicates that model $\boldsymbol{\theta}$ dominates $\boldsymbol{\theta}'$. That is, we can find a task where $\boldsymbol{\theta}'$ is worse than $\boldsymbol{\theta}$ but we can not find a task $t$ where $\ell_t(\boldsymbol{\theta}) > \ell_t(\boldsymbol{\theta}')$. For example, $\boldsymbol{\theta}$ and $\boldsymbol{\theta}'$ might have similar performance on math tasks but $\boldsymbol{\theta}$ is safer and should thus be preferred. The set of models that fulfill this criterion is called Pareto set and denoted with $\boldsymbol{\theta}_*$. Its image in the evaluation metric is called Pareto front. While Eq. 1 is theoretically appealing, as for convex $\ell_t$ any stationary point is guaranteed to be in the Pareto set (Miettinen, 1999, Theorem 1.10.10), it can quickly become too costly with large models, for example, when many $\boldsymbol{\alpha}$s yield similar solutions and the distribution of the $\boldsymbol{\alpha}$ is not uniform.

Despite its importance, not much work has been done to find good ways to set the weights for multitask finetuning. An exhaustive search over the whole $\boldsymbol{\alpha}$-space is not feasible when $T$ is large. With little guidance, arbitrary values are tried to get an idea, for example, (Fujii et al., 2024) try only two values of $\boldsymbol{\alpha}$ for continual pretraining. Sometimes heuristics are used and meta-learning approaches are also adopted, but the results are not always satisfactory, for example, see (Liu et al., 2023, Sec. 6) who report such a result for code generation models. One related work that aims to find Pareto fronts more efficiently is the one of Li et al. (2025a), who use model merging to find the performance of some $\boldsymbol{\alpha}$ combinations and then fit a quadratic model to the evaluation metric for each task which is eventually used to find Pareto fronts with existing multi-objective optimization solvers. In contrast, our method does not require such solvers, because the surrogate is automatically derived using variational learning. This makes it easy to introduce more flexible approximations beyond a quadratic one by simply choosing a more flexible posterior form. We will show this next by connecting Bayesian methods and model merging to introduce variational model merging as a new framework for estimating Pareto fronts with more accurate methods.

## 3 ESTIMATING PARETO FRONTS VIA VARIATIONAL MODEL MERGING

Our goal in this paper is to provide fast and cheap methods to produce high-quality estimates of the Pareto set. This is useful, for example, to highlight a few $\boldsymbol{\alpha}$ that can be tried out for a multitask finetuning. Model merging could be useful for this but the choice of merging strategy matters to get estimates that are diverse and cover the Pareto front well. However, no work exists that compares model merging strategies for this task or provides recipes to improve estimates. Rather, many current model merging methods focus only on the best performing weights (Don-Yehiya et al., 2023; Jiang et al., 2023; Feng et al., 2024; Stoica et al., 2024; Yang et al., 2024). We will now show that Bayesian methods provide tools to easily design merging strategies and understand their performance.

### 3.1 A BAYESIAN APPROACH TO MODEL MERGING

We view model merging as distributed Bayesian computation, where there is a natural way for weighted-merging of information distributed in different locations. Consider a multitask setup with $t$ tasks with data $\mathcal{D}_t$ and corresponding likelihoods $p(\mathcal{D}_t \,|\, \boldsymbol{\theta})$ as well as a common prior $p_0(\boldsymbol{\theta})$.[1]

The multitask posterior in this setting has a closed form solution and can simply be computed as a multiplication of posteriors raised to weights $\alpha_t > 0$ and divided by the prior:

$$p_{\boldsymbol{\alpha}}(\boldsymbol{\theta}) \;\propto\; p_0(\boldsymbol{\theta}) \prod_{t=1}^{T} p(\mathcal{D}_t|\boldsymbol{\theta})^{\alpha_t} \;\propto\; p_0(\boldsymbol{\theta})^{\gamma} \prod_{t=1}^{T} p_t(\boldsymbol{\theta})^{\alpha_t}, \qquad (3)$$

where $\gamma = 1 - \sum_t \alpha_t$. Interestingly, such posterior merging is popular in Bayesian literature, see e.g. Bayesian committee machine (Tresp, 2000) and Bayesian data fusion (Mutambara, 1998; Durrant-Whyte & Stevens, 2001) but the connection to model merging has not been explored yet.

The connection is significant: posterior merging directly allows us to exactly recover solutions of the scalarized multi-objective learning problem in Eq. 1, by choosing

$$p(\mathcal{D}_t|\boldsymbol{\theta}) \propto \exp(-\ell_t(\boldsymbol{\theta})), \qquad p_0(\boldsymbol{\theta}) \propto \exp(-\mathcal{R}_0(\boldsymbol{\theta})),$$

---

[1] The assumption of a common prior is not a necessary one and can be relaxed to using dataset specific priors.

following the generalized-Bayesian framework (Zhang, 1999; Catoni, 2007; Bissiri et al., 2016), where likelihoods are defined using general losses. With these choices, the minimizer $\boldsymbol{\theta}_{\boldsymbol{\alpha}}$ of Eq. 1 is simply the maximum-a-posterior (MAP) solution of the merged posterior $p_{\boldsymbol{\alpha}}$. That is,

$$\boldsymbol{\theta}_{\boldsymbol{\alpha}} = \arg\max_{\boldsymbol{\theta} \in \mathbb{R}^P} \log p_{\boldsymbol{\alpha}}(\boldsymbol{\theta}) = \arg\max_{\boldsymbol{\theta} \in \mathbb{R}^P} \sum_{t=1}^{T} \alpha_t \underbrace{\log p_t(\boldsymbol{\theta})}_{=\widehat{\ell}_t(\boldsymbol{\theta})} + \gamma \underbrace{\log p_0(\boldsymbol{\theta})}_{=\widehat{\mathcal{R}}_0(\boldsymbol{\theta})}. \tag{4}$$

where the $\log$ directly leads to Eq. 1. Under this lens, we can also view the loss term and regularizer term as surrogates $\widehat{\ell}_t(\boldsymbol{\theta})$ and $\widehat{\mathcal{R}}_0(\boldsymbol{\theta})$ which are used in place of the loss and regularizer of Eq. 1.

While the above is guaranteed to recover the exact solution, it can not be easily computed. The key innovation of this paper is to instead use variational approximations, which replace the distributions $p_t$ and $p_0$ by tractable approximations that can easily be computed while retaining the important property that merging methods can be obtained by taking the $\arg\max$ of the approximate posterior.

## 3.2 Variational Model Merging with Exponential-Family Surrogates

Variational learning aims to find tractable approximations of the posteriors $p_t(\boldsymbol{\theta})$ by using the following optimization problem

$$q_t(\boldsymbol{\theta}) = \arg\min_{q \in \mathcal{Q}} \ \mathbb{E}_q[\ell_t(\boldsymbol{\theta})] + \mathbb{D}_{\mathrm{KL}}[q(\boldsymbol{\theta}) \,\|\, p_0(\boldsymbol{\theta})], \tag{5}$$

where $p_0$ is again a prior and $\mathcal{Q}$ is a (sub-)class of distribution to which the optimization is restricted. We choose $\mathcal{Q}$ to be Gaussian distributions or their mixtures, because there exist optimizers of Eq. 5 that work well for deep learning (Khan & Rue, 2023). For example, the IVON optimizer (Shen et al., 2024) can estimate a (diagonal) Gaussian distribution over neural networks at a similar performance and speed as conventional optimizers like AdamW (Loshchilov & Hutter, 2019). It is even possible to use AdamW directly (Li et al., 2025b) which can also be seen as a Laplace approximation with a second-order estimate whose quality depends on the minibatch size (Khan et al., 2018).

Using variational learning, model merging schemes can be found by using the following recipe, which follows the approach of Eq. 4 but instead uses a specific family of distributions,

$$\boldsymbol{\theta}_{\boldsymbol{\alpha}} \approx \arg\max_{\boldsymbol{\theta} \in \mathbb{R}^P} \log \left[ p_0(\boldsymbol{\theta})^{\gamma} \prod_{t=1}^{T} q_t(\boldsymbol{\theta})^{\alpha_t} \right] = \arg\max_{\boldsymbol{\theta} \in \mathbb{R}^P} \sum_{t=1}^{T} \alpha_t \log q_t(\boldsymbol{\theta}) + \gamma \log p_0(\boldsymbol{\theta}), \tag{6}$$

The family of distributions then defines the surrogate that is used as follows,

$$\widehat{\ell}_t(\boldsymbol{\theta}) = -\log q_t(\boldsymbol{\theta}), \tag{7}$$

where the choice of $q_t$ and $p_0$ determine both the merging equation and quality of the estimation.

A simple choice could be to use isotropic Gaussians $q_t = \mathcal{N}(\boldsymbol{\theta} \,|\, \boldsymbol{\theta}_t, \mathbf{I})$, for example, obtained using Laplace's method around $\boldsymbol{\theta}_t$ (Khan & Rue, 2023, App. C.1), as well as a zero-mean Gaussian prior $p_0 = \mathcal{N}(\boldsymbol{\theta} \,|\, \mathbf{0}, \mathbf{I})$. Then, by direct calculation using the log of the densities the surrogates simply reduce to

$$\widehat{\ell}_t(\boldsymbol{\theta}) = \tfrac{1}{2}\|\boldsymbol{\theta} - \boldsymbol{\theta}_t\|^2 + \mathrm{const}, \qquad \widehat{\mathcal{R}}_0(\boldsymbol{\theta}) = \tfrac{1}{2}\|\boldsymbol{\theta}\|^2. \tag{8}$$

This leads to the following model merging equation which is again obtained by direct calculation

$$\widehat{\boldsymbol{\theta}}_{\boldsymbol{\alpha}}^{\mathrm{SA}} = \arg\min_{\boldsymbol{\theta} \in \mathbb{R}^P} \alpha_t \tfrac{1}{2}\|\boldsymbol{\theta} - \boldsymbol{\theta}_t\|^2 + \gamma \tfrac{1}{2}\|\boldsymbol{\theta}\|^2 = \sum_{t=1}^{T} \alpha_t \boldsymbol{\theta}_t. \tag{9}$$

Here, the last equality follows from choosing $\gamma = 1 - \sum_t \alpha_t$ and $\sum_t \alpha_t = 1$. This shows that merging by averaging (Wortsman et al., 2022) corresponds to a simple isotropic Gaussian approximations which can also be seen as a simplistic Taylor approximation where we assume $\nabla \ell_t(\boldsymbol{\theta}_t)$ to be zero (due to local optimality) and the Hessian $\nabla^2 \ell_t(\boldsymbol{\theta}_t)$ is set to identity,

$$\begin{aligned} \ell_t(\boldsymbol{\theta}) &\approx \ell_t(\boldsymbol{\theta}_t) + \nabla \ell_t(\boldsymbol{\theta}_t)^{\top}(\boldsymbol{\theta} - \boldsymbol{\theta}_t) + \tfrac{1}{2}(\boldsymbol{\theta} - \boldsymbol{\theta}_t)^{\top} \nabla^2 \ell_t(\boldsymbol{\theta}_t)(\boldsymbol{\theta} - \boldsymbol{\theta}_t) \\ &\approx \ell_t(\boldsymbol{\theta}_t) + \tfrac{1}{2}\|\boldsymbol{\theta} - \boldsymbol{\theta}_t\|^2. \end{aligned} \tag{10}$$

---

**Algorithm 1** Fast and cheap multitask Pareto estimates via mixture of Gaussian merging

---

**Require:** $K$ different Gaussians for each of the $T$ tasks $\mathcal{N}(\boldsymbol{\theta} \,|\, \boldsymbol{\theta}_{tk}, \mathrm{diag}(1/\mathbf{h}_{tk}))$.

1: **for** all $\boldsymbol{\alpha}$ values in the preview **do**
2:     Initialize $\widehat{\boldsymbol{\theta}}_{\boldsymbol{\alpha}}$ and set $\pi_k = 1/K$ for all $k$.
3:     **while** not converged **do**
4:        For all $t, k$: compute $p_{tk} = \pi_k \mathcal{N}(\widehat{\boldsymbol{\theta}}_{\boldsymbol{\alpha}} \,|\, \boldsymbol{\theta}_{tk}, \mathrm{diag}(1/\mathbf{h}_{tk}))$; normalize $\hat{\pi}_{tk} \leftarrow \frac{p_{tk}}{\sum_{k'} p_{tk'}}$
5:        $\mathbf{h}_{\boldsymbol{\alpha}} \leftarrow \sum_{t,k} \hat{\pi}_{tk} \alpha_t \mathbf{h}_{tk}$
6:        $\widehat{\boldsymbol{\theta}}_{\boldsymbol{\alpha}} \leftarrow (\sum_{t,k} \hat{\pi}_{tk} \alpha_t \, \mathbf{h}_{tk} \boldsymbol{\theta}_{tk})/\mathbf{h}_{\boldsymbol{\alpha}}$
7:     **end while**
8: **end for**

---

The surrogates $\widehat{\ell}_t$ are tight at only one point and their inaccuracy increases as we move away from it. Model merging can be seen as using $\sum_t \alpha_t \widehat{\ell}_t$ (along with regularizer $\widehat{\mathcal{R}}_0$) as a proxy to estimate the scalarized multi-objective problem $\sum_t \alpha_t \ell_t$. However, when using a wide range of $\boldsymbol{\alpha}$ values, these inaccuracies can lead to poor estimates in some regions of the Pareto front. These errors can lead to poor estimates on regions away from the extremes, when the $\boldsymbol{\alpha}$ gives all tasks similar weightings.

Many other merging strategies can also be seen under this lens by defining suitable variational approximations and priors, and surrogates. For example, for task arithmetic (Ilharco et al., 2023; Ortiz-Jimenez et al., 2023), we would only change the prior to a non-zero-mean Gaussian $p_0(\boldsymbol{\theta}) = \mathcal{N}(\boldsymbol{\theta} \,|\, \boldsymbol{\theta}_0, \mathbf{I})$, where $\boldsymbol{\theta}_0 \in \mathbb{R}^P$ could be a pretrained LLM, for example. Then, the surrogate regularizer is $\widehat{\mathcal{R}}_0(\boldsymbol{\theta}) = \frac{1}{2} \|\boldsymbol{\theta} - \boldsymbol{\theta}_0\|^2$ which gives rise to the *task vectors* $\tau_t = \boldsymbol{\theta}_t - \boldsymbol{\theta}_0$ and merged model

$$\widehat{\boldsymbol{\theta}}_{\boldsymbol{\alpha}}^{\mathrm{SA}} = \sum_{t=1}^{T} \boldsymbol{\theta}_0 + \alpha_t(\boldsymbol{\theta}_t - \boldsymbol{\theta}_0). \tag{11}$$

More precise approximations are obtained by using more expressive posteriors. For example, a full-Gaussian posterior would also consider variance information and include isotropic Gaussians as a special case. In a full-Gaussian posterior the Hessian $\mathbf{H}_t$ can be used to encode second-order information, for example, again by using Laplace's method. Then, by using $q_t(\boldsymbol{\theta}) = \mathcal{N}(\boldsymbol{\theta} \,|\, \boldsymbol{\theta}_t, \mathbf{H}_t^{-1})$, we get the surrogate $\widehat{\ell}_t(\boldsymbol{\theta}) = \|\boldsymbol{\theta} - \boldsymbol{\theta}_t\|_{\mathbf{H}_t}^2$ which in turn leads to a Hessian-based merging,

$$\widehat{\boldsymbol{\theta}}_{\boldsymbol{\alpha}}^{\mathrm{Hess}} = \Big( \sum_t \alpha_t \mathbf{H}_t \Big)^{-1} \sum_t \alpha_t \mathbf{H}_t \boldsymbol{\theta}_t, \tag{12}$$

where $p_0(\boldsymbol{\theta}) = \mathcal{N}(\boldsymbol{\theta} \,|\, 0, \mathbf{I})$, $\sum_t \alpha_t = 1$, and approximating the Hessians by the Fishers $\mathbf{F}_t$ recovers Fisher-weighted Averaging Matena & Raffel (2022).

Choosing a non-zero-mean prior, as we did to derive Task Arithmetic but with Hessian $(\mathbf{H}_0 + \mathbf{H}_t)$ recovers the method from Daheim et al. (2024). This formulation also recovers the closed-form Pareto solution for convex quadratic multi-objective problems (Sheftel et al., 2013, Appendix S1).

In general, variational model merging allows us to use any exponential-family distribution for $q_t$ and $\widehat{\ell}_t$. Then, we are guaranteed to have a closed-form solution to Eq. 6, because

$$q_t(\boldsymbol{\theta}) \propto e^{\boldsymbol{\lambda}_t^\top \mathbf{T}(\boldsymbol{\theta})} \quad \Longrightarrow \quad \widehat{\ell}_t(\boldsymbol{\theta}) = -\boldsymbol{\lambda}_t^\top \mathbf{T}(\boldsymbol{\theta}) + \mathrm{const.}$$

where we denote sufficient statistics by $\mathbf{T}(\boldsymbol{\theta})$ and the natural parameter by $\boldsymbol{\lambda}_t$. The merging has a closed form solution because the minimizer of the weighted sum $\sum_t \alpha_t \widehat{\ell}_t$ is equivalent to the MAP of an exponential family distribution which is always available in closed-form. This is explained in further detail in App. A.2. The surrogates not only take flexible forms, but also are more globally accurate. This is because they are obtained by solving Eq. 5 which is equivalent to minimizing the KL divergence to the exact posterior $p_t$. This ensures that the surrogates are accurate not only locally at $\boldsymbol{\theta}_t$ but also globally in regions where $q_t$ has high probability mass (Opper & Archambeau, 2009).

### 3.3 IMPROVED MERGING VIA MIXTURES OF EXPONENTIAL-FAMILIES

Here, taking advantage of the variational model merging framework, we derive a new strategy using mixture of exponential-family distributions which provide more expressive posteriors and therefore

even more accurate surrogates. For simplicity, we assume no regularizer and that $\sum_t \alpha_t = 1$. While mode finding for mixtures is still tractable, it requires an iterative expectation-maximization (EM) procedure which should still be cheap if it converges within few steps. We assume that the $k$'th mixture component is an EF with natural parameter $\boldsymbol{\lambda}_{tk}$. Each component is weighted by $\pi_k > 0$ and $\sum_k \pi_k = 1$. Then, the posterior and surrogate take the following form:

$$q_t \propto \sum_{k=1}^{K} \underbrace{\pi_k e^{\boldsymbol{\lambda}_{tk}^{\top} \mathbf{T}(\boldsymbol{\theta})}}_{\propto p_{tk}(\boldsymbol{\theta})} \quad \implies \quad \widehat{\ell}_t(\boldsymbol{\theta}) = -\log \sum_{k=1}^{K} \pi_k e^{\boldsymbol{\lambda}_{tk}^{\top} \mathbf{T}(\boldsymbol{\theta})}. \tag{13}$$

Clearly, the surrogate is much more expressive than the quadratic ones from previous model merging strategies, because they are contained as special cases. Despite the non-concavity of the objective, we can maximize it using an iterative Expectation-Maximization (EM) approach where each step has a closed-form solution. A detailed derivation is in App. A.4. As a special case, consider a mixture-of-Gaussians (MoG) posterior where the updates take the following form similarly to Eq. 12:

$$\boldsymbol{\theta}^{(i+1)} \leftarrow (\mathbf{H}_{\boldsymbol{\alpha}}^{(i)})^{-1} \sum_{t,k} \hat{\pi}_{tk}^{(i)} \alpha_t \mathbf{H}_{tk} \boldsymbol{\theta}_{tk}, \text{ where } \mathbf{H}_{\boldsymbol{\alpha}}^{(i)} = \sum_{t,k} \hat{\pi}_{tk}^{(i)} \alpha_t \mathbf{H}_{tk}. \tag{14}$$

The main difference is that each component is now weighted by $\hat{\pi}_{tk}^{(i)} \propto \pi_k \mathcal{N}(\boldsymbol{\theta}^{(i)} \mid \boldsymbol{\theta}_{tk}, \mathbf{H}_{tk}^{-1})$, normalized over $k$. This update generalizes the fixed-point algorithm of Carreira-Perpiñán (2000, Section 5) which was proposed to find the modes of Gaussian mixtures.

## 3.4 PROOF THAT MORE FLEXIBLE POSTERIORS YIELD BETTER MERGING

We now prove that more flexible posteriors yield better merging methods. We use a result by Khan (2025) that gives an alternative *dual* form for solutions of the variational objective in terms of local surrogates. Specifically, consider the following posterior obtained by training with a weighted loss,

$$q_{\boldsymbol{\alpha}}^* = \arg\min_{q \in \mathcal{Q}} \sum_{t=1}^{T} \alpha_t \mathbb{E}_q[\ell_t(\boldsymbol{\theta})] + \mathbb{D}_{\mathrm{KL}}[q(\boldsymbol{\theta}) \,\|\, p_0(\boldsymbol{\theta})].$$

This is the gold standard that we want to estimate by using variational model merging. Khan (2025, Eq. 3) shows that the posterior can be written in a dual form in terms of surrogates $\widehat{\ell}_t^*$,

$$q_{\boldsymbol{\alpha}}^* \propto p_0 \prod_{t=1}^{T} \exp\left(-\widehat{\ell}_t^*(\boldsymbol{\theta})\right)^{\alpha_t}.$$

The surrogates are similarly defined as used in this paper, but we spare these details since the exact form is less important for the result we prove. What is important to note is that the surrogates $\widehat{\ell}_t^*$ are optimal in the sense that multiplying them together yields the desired $q_{\boldsymbol{\alpha}}^*$ exactly.

The dual form allows quantifying the divergence between $q_{\boldsymbol{\alpha}}^*$ and a merged posterior, say,

$$q_{\boldsymbol{\alpha}} = p_0^{\gamma} \prod_{t=1}^{T} \exp(-\widehat{\ell}_t(\boldsymbol{\theta}))^{\alpha_t}.$$

Then, we can write the KL divergence between $q_{\boldsymbol{\alpha}}^*$ and $q_{\boldsymbol{\alpha}}$ as follows. We drop $(\boldsymbol{\theta})$ for succinctness.

$$\mathbb{D}_{\mathrm{KL}}[q_{\boldsymbol{\alpha}}^*(\boldsymbol{\theta}) \,\|\, q_{\boldsymbol{\alpha}}(\boldsymbol{\theta})] = \mathbb{E}_{q_{\boldsymbol{\alpha}}^*} \left[ \log \frac{p_0 \prod_{t=1}^{T} \exp\left(-\widehat{\ell}_t^*\right)^{\alpha_t}}{p_0^{\gamma} \prod_{t=1}^{T} \exp(-\widehat{\ell}_t)^{\alpha_t}} \right]$$

$$= \sum_{t=1}^{T} \alpha_t \mathbb{E}_{q_{\boldsymbol{\alpha}}^*} [(\widehat{\ell}_t - \widehat{\ell}_t^*)] + (1 - \gamma) \mathbb{E}_{q_{\boldsymbol{\alpha}}^*} [\log p_0].$$

The second term is less important here, but the first term directly captures the approximation error made by the surrogates. Essentially, smaller values of the sum $\widehat{\ell}_t - \widehat{\ell}_t^*$ imply better merging.

From this result, it directly follows that more flexible posteriors yield better merging. This is because more flexible posteriors necessarily lead to a better posterior approximation and a better surrogate. For example, isotropic Gaussians lead to linear surrogates while full Gaussians lead to quadratic

Table 1: We report the maximum accuracy/BLEU obtained via different variational merging strategies; corresponding $\boldsymbol{\alpha}$ values are in Table 2. With each score, we also show (in grey) the best score of multitask finetuning obtained using one of the best five weights found with merging. Finally, the maximum scores for multitask finetuning over all $\boldsymbol{\alpha}$ values are shown in the last row. We see a consistent trend that the performance improves as we use better posteriors. For example, for Fig. 8, the accuracy increases from 76.9% to 87.2% which is close to the 90.4% for multitask finetuning.

| Model | Logistic | | ResNet-20 | ViT-B/32 | | RoBERTa | GEMMA-2B |
|---|---|---|---|---|---|---|---|
| | Fig. 8 | Fig. 7 | Fig. 2 | Fig. 11 | Fig. 3 | Fig. 4 | Fig. 5 |
| Simple Merging | 76.9% 90.2% | 71.3% 90.7% | 63.2% 67.4% | 86.0% 97.1% | 79.2% 84.1% | 93.5% 95.2% | 25.9 26.7 |
| Hessian Weighted | 80.3% 89.4% | 78.5% 90.6% | 63.3% 65.9% | 88.3% 97.2% | 80.0% 84.1% | 93.6% 95.5% | 25.9 26.7 |
| Mixture Weighted | 87.2% 90.4% | 83.2% 90.5% | 64.6% 68.1% | 88.1% 97.2% | 80.5% 83.5% | 94.1% 95.4% | 26.2 26.6 |
| Multitask Finetuning | 90.4% | 90.7% | 68.1% | 97.3% | 84.4% | 95.5% | 26.7 |

surrogates. Therefore, we expect a smaller KL divergence. More precisely, consider two merged solution $q_{\boldsymbol{\alpha}}^1$ and $q_{\boldsymbol{\alpha}}^2$ constructed using two different variational families $\mathcal{Q}^1$ and $\mathcal{Q}^2$ respectively. Now suppose that the first is a strict subset of the latter, that is, $\mathcal{Q}^1 \subset \mathcal{Q}^2$, then we also know that $q_{\boldsymbol{\alpha}}^2$ is never worse than $q_{\boldsymbol{\alpha}}^1$, and vice versa. More formally,

$$\mathbb{D}_{\mathrm{KL}}[q_{\boldsymbol{\alpha}}^*(\boldsymbol{\theta}) \,\|\, q_{\boldsymbol{\alpha}}^2(\boldsymbol{\theta})] \leq \mathbb{D}_{\mathrm{KL}}[q_{\boldsymbol{\alpha}}^*(\boldsymbol{\theta}) \,\|\, q_{\boldsymbol{\alpha}}^1(\boldsymbol{\theta})] \quad \Leftrightarrow \quad \sum_{t=1}^T \alpha_t \mathbb{E}_{q_{\boldsymbol{\alpha}}^*}\left[\widehat{\ell}_t^2 - \widehat{\ell}_t^*\right] \leq \sum_{t=1}^T \alpha_t \mathbb{E}_{q_{\boldsymbol{\alpha}}^*}\left[\widehat{\ell}_t^1 - \widehat{\ell}_t^*\right].$$

This proves that more flexible posteriors imply better or at least as good merging. This holds for all $\boldsymbol{\alpha}$, therefore we can also conclude that Pareto-front estimates would also be better.

### 3.5 Algorithms for Fast Multitask Finetuning Pareto estimates

Our methods to generate Pareto front estimates follow four steps: (1) Finetune $T$ models (denoted by $\boldsymbol{\theta}_t$) each separately over their own task $\ell_t(\boldsymbol{\theta})$, (2) Use Bayesian learning to build surrogates $\ell_t \approx \widehat{\ell}_t$ by using $q_t(\boldsymbol{\theta})$, (3) Create Pareto estimates by evaluating $\sum_t \alpha_t \widehat{\ell}_t$ for many $\boldsymbol{\alpha}$ values. (4) Use good $\boldsymbol{\alpha}$'s to finetune a multitask model. Next, we list four versions with different algorithms.

1. ADAMW-SG: We train each task using AdamW (Loshchilov & Hutter, 2019) to get $\boldsymbol{\theta}_t$ and use it to build the Laplace posterior $q_t(\boldsymbol{\theta}) = \mathcal{N}(\boldsymbol{\theta} \,|\, \boldsymbol{\theta}_t, \mathbf{H}_t^{-1})$. The Hessian is fixed to a diagonal matrix with the diagonal set to squared gradients, which are computed by one extra pass through the data. Pareto estimates are then computed using Eq. 12 for all $\boldsymbol{\alpha}$.

2. IVON-HESS: We train each task using the variational learning method IVON (Shen et al., 2024) which yields a Gaussian with diagonal covariance, similarly to AdamW-SG. The advantage here is that no additional pass through the data is required to compute the diagonal Hessian. Again, Pareto estimates are computed by using Eq. 12.

3. MULTIIVON-HESS. For each task, we train using multiple runs of IVON and use them to construct a mixture-of-Gaussians posterior. The cost is $K$ times the cost of IVON-Hess where $K$ is the number of IVON runs. Pareto estimates are obtained using Alg. 1.

4. SOUPIVON-HESS. For each task, we train $L < K$ runs of IVON and use multiple checkpoints from each run to construct a mixture-of-Gaussians posterior. The cost is only $\mathcal{O}(L \cdot T)$ at training time. Pareto estimates are obtained using Alg. 1.

For MultiIVON-Hess and SoupIVON-Hess, we only run 5-10 iterations of Alg. 1 with $K$ set in the range of 3 to 30 components. MultiIVON-Hess is still practical but, compared to other methods, can have larger training overhead for large models due to having to train 3-30 models for each task.

## 4 Experiments & Results

We compare the Pareto fronts obtained with multitask finetuning and (variational) model merging on image classification using ResNets (Sec. 4.1) and ViTs (Sec. 4.2), text classification with masked language models (Sec. 4.3), and machine translation using LLMs (Sec. 4.4). In App. C.2 we show results for multitask learning on logistic regression with MNIST dataset. An overview of our results

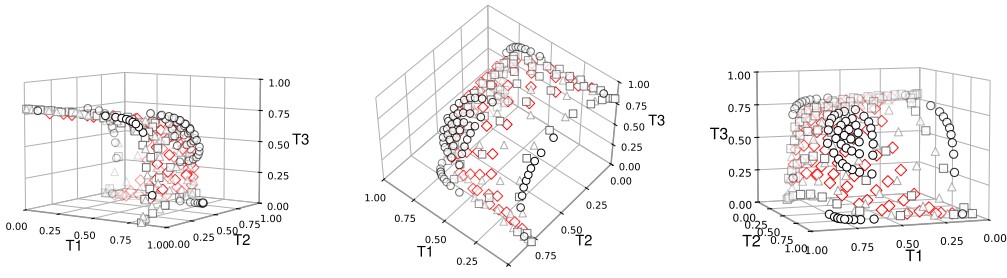

Figure 2: Results on using ResNet-20 on CIFAR-10 with three tasks constructed from different sets of classes. Notably, the mixture (◇) finds better Pareto Fronts than Hessian-Weighted merging (□) and Simple Merging (△). Their expressiveness produces diverse models that also cover trade-off regions that even multitask finetuning (○) which several $\alpha$ weightings could not reach.

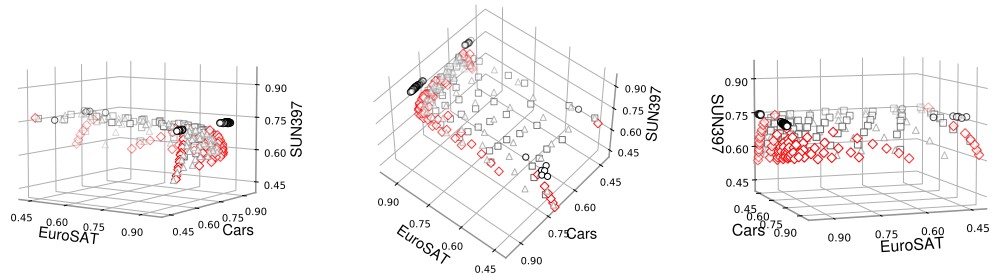

Figure 3: Results using ViT-B/32 on EuroSAT, Cars, SUN397. Multitask finetuning (○) results in three clusters. Mixture of Gaussian merging (◇) appears less uniform than Hessian-Weighted Merging (□) and Simple merging (△) and is closing in on the clusters found by multitask training.

is shown in Table 1: we find that more expressive posteriors lead to better model merging results and can also lead to better results when the $\alpha$s found with them are used for multitask finetuning.

## 4.1 IMAGE CLASSIFICATION ON CIFAR-10

Here, we pretrain ResNet-20 (He et al., 2016) with 260k parameters on a subset of CIFAR-10 and then finetune it on the tasks: (1) airplane, car, ship, truck; (2) cat, dog; (3) bird, deer, frog, horse. We compare Hessian-Weighted and Mixture-Weighted Merging, both using IVON, to Simple Merging and multitask training. Results are shown in Fig. 2 and again show that better posterior approximations yield better estimates. We find that the approximated Pareto front found by more expressive posteriors, in particular Mixture-Weighted Merging, comes closer to the one of multitask training but also covers trade-offs that simple scalarization could not find, where multitask finetuning is used to try several $\alpha$s.

## 4.2 VISION TRANSFORMERS

Next, we experiment with ViT-B/32 models based on CLIP (Radford et al., 2021) for image classification. First, we use EuroSAT (Helber et al., 2019), Stanford Cars (Krause et al., 2013) and SUN397 (Xiao et al., 2010). Then, we use GTSRB (Houben et al., 2013), RESISC45 (Cheng et al., 2017) and SVHN (Netzer et al., 2011); We compare Simple Merging to Hessian-Weighted Merging (ADAMW-SG), and Mixture-Weighted Merging with SG Hessian and $K = 10$. Further details can be found in App. B.4. We use a grid with spacing 0.05 for each $\alpha_t$.

The results for the first set of tasks are shown in Fig. 3 and results the second task are shown in Fig. 11 with similar conclusions. In both cases we find that multitask finetuning mostly finds three clusters

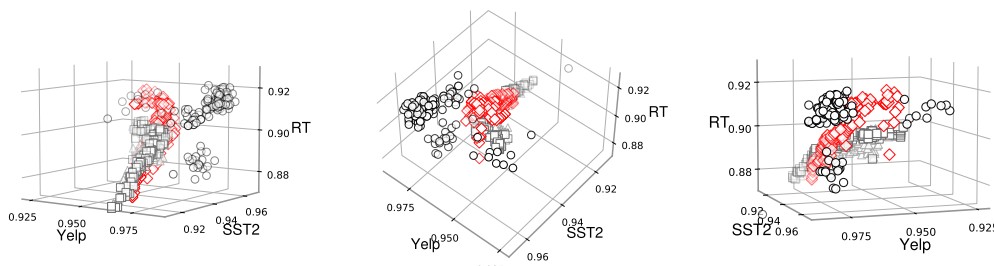

Figure 4: Pareto front estimates for RoBERTA finetuned on three sentiment analysis tasks. Mixture-Weighted variational merging (◇) produces better estimates than Hessian-weighted (□) and Simple merging (△), and finds models that lie in between the two clusters found by multitask finetuning (○).

that roughly correspond to one of the datasets performing very well. Mixture-Weighted Merging appears to come closer to these clusters than Hessian-Weighted Merging which in turn appears to perform better than Simple Merging. Especially the Pareto front of Simple Merging is mostly dominated by the Pareto front of other merging methods. In addition, our merging methods seem to find trade-offs that are not immediately found by scalarization and multitask training with a few $\alpha$. Finally, training times underscore the usefulness of our methods: joint training for one weighted combination takes around 51 minutes while merging takes only seconds, in addition to 14-17 minutes for each finetuning on separate tasks but this only has to be performed once.

### 4.3 MASKED LANGUAGE MODELS

In this section, we show results for multitask finetuning masked language models for text sentiment classification. We follow Daheim et al. (2024) and train RoBERTa (Liu et al., 2019) first on IMDB (Maas et al., 2011). Then, we finetune on Rotten Tomatoes (RT) (Pang & Lee, 2005), SST-2 (Socher et al., 2013), and Yelp (Zhang et al., 2015), and merge the resulting models. We merge all task models using Simple Merging and Hessian-Weighted (ADAMW-SG) and mixture of Gaussians with squared gradient Hessians. The resulting Pareto fronts are shown in Fig. 4 and the full shape of the results is shown in Fig. 5a. We find that the Pareto front of Mixture-Weighted merging again comes closer to the multitask finetuning Pareto than Hessian-Weighted Merging (AdamW-SG) and Simple Merging. For the overall shapes, we find that it also matches the shape of the multitask finetuning run best. In general, we find that the shape approximation gets better and better as the posterior becomes more flexible. Note that we only use mixtures of $K = 3$ components here and that using more components might improve results further.

### 4.4 MACHINE TRANSLATION WITH FINETUNED LLMS

Finally, we show that our methods are also practical for billion-parameter-scale LLMs and when finetuned using parameter-efficient finetuning strategies such as LoRA. In particular, we merge two GEMMA-2B-it (Gemma Team, 2024b) models finetuned on IWSLT2017 (Cettolo et al., 2017) de-en and fr-en, respectively, and compare them to training jointly on both language pairs as well as Mixture-Weighted merging with $K = 3$. Here, we use IVON-HESS. Details about the experimental set-up are in App. B.6. Results are shown in Fig. 5b. There, we find that Simple Merging does not always match the shape of the Pareto front of multitask finetuning, especially around $\alpha = 0.4$. Using Hessian-Weighted merging improves this. Mixture-Weighted merging provides overall the best results but does not always match the shape well. Overall, this shows that the our method also scales to larger models and datasets and parameter-efficient finetuning. One run of multitask finetuning takes around 17h while merging takes just 1 minute (plus 8-9h for finetuning on each task separately). The EM algorithm from Alg. 1 is also fast, as it is only run on the ca. 1M LoRA parameters.

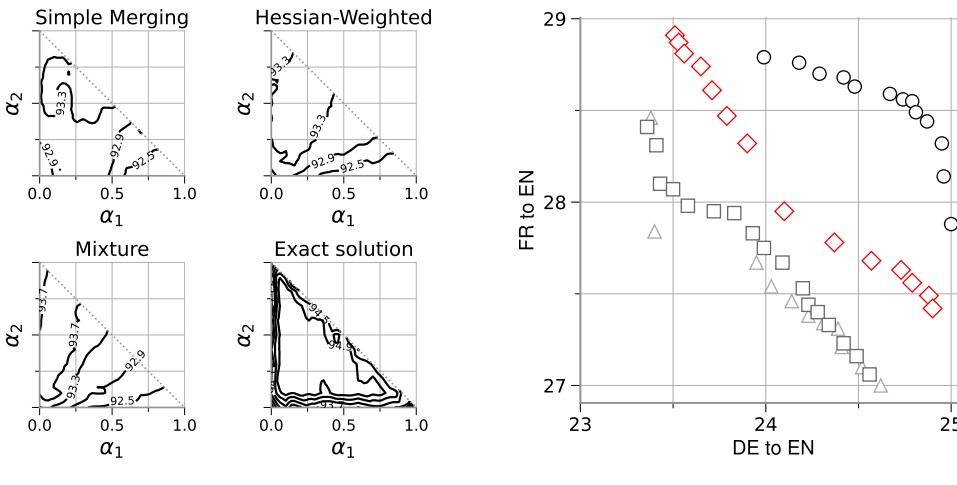

(a) RoBERTa for Sentiment Classification   (b) GEMMA-2B for Machine Translation

Figure 5: In Fig. 5a we show the full shape of trade-offs for different $\alpha$s for all merging methods and multitask finetuning beyond only the Pareto front. This can be interesting to understand general trade-offs between tasks. We find that better posteriors also lead to broader well-performing regions that capture the overall shape of task trade-offs better. In Fig. 5b we compare mixture of Gaussian merging ($\diamond$) against Hessian-Weighted ($\square$) and Simple merging ($\triangle$) as well as multitask finetuning ($\circ$) of GEMMA-2B on IWSLT2017 de-en and fr-en translation tasks. We find that more flexible posteriors generally give better merging results closer to multitask training.

## 5 Limitations

One limitation of our work is that exact Bayesian posteriors can in general not be obtained efficiently or at all due to a large computational burden. Due to this, the posteriors can only be approximated which leads to gaps between the performance of the merged models and exact solution. Similarly, the Hessians required by some of our methods can not be calculated and stored exactly for larger problems that typically involve large neural networks due to the quadratic scaling in parameter size. Therefore, again approximations are required and these introduce performance losses. In particular, throughout this paper, we frequently resort to diagonal approximations of the Hessians.

## 6 Conclusion

Multitask finetuning is a crucial ingredient in many neural network training recipes but good weightings between tasks are hard and expensive to find. Here, we propose to aid the search for such weightings with Pareto estimates obtained from model merging, where single task models can be reused for many weight combinations. We show that model merging strategies can be derived using a Bayesian framework by defining suitable surrogate losses to the multitask objective for exponential-family-based distributions. We use this to outline various merging strategies, including a new mixture-based algorithm for improved model merging. Along various experiments including image classification with Vision Transformers and machine translation with LLMs we show that model merging can effectively be used to get estimates and choose multitask finetuning weightings. Flexible model merging can improve the estimate quality, but also increase the cost due to the requirement of computing Hessians and using multiple checkpoints. However, strategies to estimate Hessians during training, such as, variational learning, can be used to mitigate this.

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

## A  DERIVATIONS

### A.1  CLOSED-FORM EXPRESSION FOR PARETO-OPTIMALITY IN VARIATIONAL BAYES

In Eq. 4 we connected the minimizer of $\sum_t \alpha_t \ell_t + \mathcal{R}_0$ to Bayesian learning, however we can also consider a more general problem, where we have $T$ functions $f_t$, so the problem is:

$$\boldsymbol{\theta}^* = \arg\min_{\boldsymbol{\theta}} [f_1(\boldsymbol{\theta}), f_2(\boldsymbol{\theta}), \cdots, f_M(\boldsymbol{\theta})]. \tag{15}$$

As mentioned in Sec. 2, this can be scalarized to solve for a particular set of $\boldsymbol{\alpha}$, and its minimizer is Pareto optimal if $f_t$ are convex, or weakly Pareto optimal in the general case.

$$\boldsymbol{\theta}_{\boldsymbol{\alpha}} = \arg\min_{\boldsymbol{\theta}} \sum_t \alpha_t f_t(\boldsymbol{\theta}). \tag{16}$$

If we each posterior as $p_t \propto \exp(-f_t(\boldsymbol{\theta}))$ we see the connection to multi-objective optimization.

$$\boldsymbol{\theta}_{\boldsymbol{\alpha}} = \arg\max_{\boldsymbol{\theta}} \sum_t \alpha_t(\log \text{lik}_t + \log p_0)$$

$$= \arg\max_{\boldsymbol{\theta}} \log \prod_t (\text{lik}_t \times p_0)^{\alpha_t} = \log \prod_t p_t^{\alpha_t} + \text{const.} = \log p_{\boldsymbol{\alpha}} + \text{const.} \tag{17}$$

Therefore, the mode of the joint posterior $p_{\boldsymbol{\alpha}}$ for a given $\boldsymbol{\alpha}$ recovers the Pareto solution $\boldsymbol{\theta}_{\boldsymbol{\alpha}}$. The quality of the solution will depend in our choices on how $p_t$ is approximated.

### A.2  CLOSED-FORM EXPRESSION FOR MAP OF EXPONENTIAL-FAMILY DISTRIBUTION

It is clear that minimizing $\sum_t \alpha_t \widehat{\ell}_t$ is equivalent finding MAP of $\prod_t q_t(\boldsymbol{\theta})^{\alpha_t}$. Let us denote it by $p_{\boldsymbol{\alpha}}$. The mode of $p_{\boldsymbol{\alpha}}$ is available in closed-form because we can always rewrite the posterior in an exponential form that allows us to compute the max. This is shown below where we first rewrite the posterior with a log-partition function $A(\boldsymbol{\theta})$ and an alternate sufficient statistic $\mathbf{t}(\boldsymbol{\lambda}_{\alpha})$, and then simply take the derivative to get a closed-form expression for the Pareto solution,

$$p_{\boldsymbol{\alpha}} \propto \exp(\boldsymbol{\theta}^\top \mathbf{t}(\boldsymbol{\lambda}_{\alpha}) - A(\boldsymbol{\theta})) \implies \boldsymbol{\theta}_{\boldsymbol{\alpha}} = \arg\max_{\boldsymbol{\theta}} \log p_{\alpha}(\boldsymbol{\theta}) = \nabla A^{-1}(\mathbf{t}(\boldsymbol{\lambda}_{\alpha})). \tag{18}$$

The alternate form is essentially following the form of the conjugate prior (see Bishop (1995, Eq. 2.229)) written to match the form of the likelihood. For instance, in the Gaussian case,

$$p_{\boldsymbol{\alpha}} \propto \exp(\underbrace{\boldsymbol{\theta}^\top \mathbf{H}_{\boldsymbol{\alpha}} \mathbf{m}_{\boldsymbol{\alpha}}}_{\mathbf{t}(\boldsymbol{\lambda}_\alpha)} - \underbrace{\tfrac{1}{2}\boldsymbol{\theta}^\top \mathbf{H}_{\boldsymbol{\alpha}} \boldsymbol{\theta}}_{A(\boldsymbol{\theta})}) \quad \implies \quad \underbrace{\mathbf{H}_{\boldsymbol{\alpha}}(\boldsymbol{\theta}_{\boldsymbol{\alpha}})}_{\nabla A(\boldsymbol{\theta}_\alpha)} = \underbrace{\mathbf{H}_{\boldsymbol{\alpha}} \mathbf{m}_{\boldsymbol{\alpha}}}_{\mathbf{t}(\boldsymbol{\lambda}_\alpha)} \quad \implies \quad \boldsymbol{\theta}_{\boldsymbol{\alpha}} = \mathbf{m}_{\boldsymbol{\alpha}}$$

Solutions of variational objectives always have this form (Khan & Rue, 2023, Sec. 5), while convexity of $A(\boldsymbol{\theta})$ ensures that the mode always exists and can be easily found without retraining on individual objectives. In summary, we can always get a closed-form expression as follows

1. Compute $\boldsymbol{\lambda}_t$ of individual objectives.
2. Aggregate them $\boldsymbol{\lambda}_\alpha = \sum_t \alpha_t \boldsymbol{\lambda}_t$.
3. Compute $\boldsymbol{\theta}_{\boldsymbol{\alpha}}$ using Eq. 18.

### A.3 CLOSED FORM SOLUTION FOR THE BETA-BERNOULLI MODEL

To illustrate the process, we discuss another example of the Beta-Bernoulli model to model coin-flips $y_t \in \{0, 1\}$ with probability $\pi$,

$$p(y_t \,|\, \pi) \propto \pi^{y_t}(1-\pi)^{1-y_t}, \quad p(\pi) \propto \pi^{a_0-1}(1-\pi)^{b_0-1}.$$

The unknown is then modeled as $\theta = \log(\pi/(1-\pi))$, to get the posterior which is also a Beta distribution with parameters

$$a_t = y_t + a_0, \quad b_t = 1 - y_t + b_0, \quad p(\theta \,|\, \mathcal{D}_t) \propto \pi^{a_t-1}(1-\pi)^{b_t-1} \propto \exp(\boldsymbol{\lambda}_t^\top \mathbf{T}(\theta))$$

where $\mathbf{T}(\theta) = (\log \pi, \log(1-\pi))$ and $\boldsymbol{\lambda}_t = (a_t - 1, b_t - 1)$. With this, the aggregate posterior $p_\alpha$ is a Beta distribution natural parameter $\boldsymbol{\lambda}_\alpha = (a_\alpha - 1, b_\alpha - 1)$ where $a_\alpha$ and $b_\alpha$ are simply a weight average of $a_t$ and $b_t$ respectively. To get the maximum of $p_\alpha$, we write it in an exponential form,

$$p_\alpha(\theta) \propto \exp(\theta \underbrace{a_\alpha - 1}_{t(\boldsymbol{\lambda}_\alpha)} - \underbrace{(a_\alpha + b_\alpha - 2)\log(1 + e^\theta)}_{A(\theta)}) \quad \implies \quad \underbrace{\frac{a_\alpha + b_\alpha - 2}{1 + e^{-\theta_{\text{PO}}}}}_{\nabla A(\boldsymbol{\theta}_{\text{PO}})} = \underbrace{a_\alpha - 1}_{\mathbf{t}(\boldsymbol{\lambda}_\alpha)},$$

from which we get $\pi_{\text{PO}} = (a_\alpha - 1)/(a_\alpha + b_\alpha - 2)$.

### A.4 DERIVATION OF THE EM ALGORITHM FOR MIXTURE POSTERIORS

To do so, we use the EM algorithm by viewing the summation over $k$ in Eq. 13 as marginalization over a discrete variable $z_k \in \{1, 2, \dots, K\}$ of the joint $p(\boldsymbol{\theta}, z_t = k) = p_{tk}(\boldsymbol{\theta})$. Then, given parameters $\boldsymbol{\theta}^{(i)}$ at each iteration $i$, we maximize the EM lower bound. The posterior over $z_k$ is

$$p(z_t = k \,|\, \boldsymbol{\theta}^{(i)}) = \hat{\pi}_{tk}^{(i)} = \frac{p_{tk}(\boldsymbol{\theta}^{(i)})}{\sum_{k'} p_{tk'}(\boldsymbol{\theta}^{(i)})}.$$

Using this, we can write the following lower bound,

$$\sum_{t=1}^T \alpha_t \log \left( \sum_{k=1}^K \frac{p_{tk}(\boldsymbol{\theta})}{\hat{\pi}_{tk}^{(i)}} \hat{\pi}_{tk}^{(i)} \right) \geq \sum_{t=1}^T \sum_{k=1}^K \alpha_t \hat{\pi}_{tk}^{(i)} \log p_{tk}(\boldsymbol{\theta}) + \mathbf{c} = \underbrace{\sum_{t=1}^T \sum_{k=1}^K \alpha_t \hat{\pi}_{tk}^{(i)} \boldsymbol{\lambda}_{tk}^\top}_{=\boldsymbol{\lambda}_\alpha^{(i)}} \mathbf{T}(\boldsymbol{\theta}) + \mathbf{c}.$$

The above corresponds to the log of an exponential family (denoted by $p_\alpha^{(i)}$) with natural parameter $\boldsymbol{\lambda}_\alpha^{(i)}$, which gives us the following iterative procedure:

$$\boldsymbol{\theta}^{(i+1)} \leftarrow \arg\max_{\boldsymbol{\theta}} \sum_{t=1}^T \sum_{k=1}^K \hat{\pi}_{tk}^{(i)} \alpha_t \boldsymbol{\lambda}_{tk}^\top \mathbf{T}(\boldsymbol{\theta}), \tag{19}$$

where we use the posterior $\hat{\pi}_{tk}^{(i)} = p(z_t = k \,|\, \boldsymbol{\theta}^{(i)}) \propto p_{tk}(\boldsymbol{\theta}^{(i)})$ which is obtained by normalizing over $k$. The iterates $\boldsymbol{\theta}^{(i)}$ converge to a local maximum which provides a solution for $\hat{\boldsymbol{\theta}}_\alpha$. For $K = 1$, the algorithm reduces to the exponential-family case.

## B  EXPERIMENTAL SETUP

### B.1  ILLUSTRATIVE 2D EXAMPLE

The individual functions in Fig. 1b are of the form $\ell_t(\boldsymbol{\theta}) = \log\left(\sum_{i=1}^{N} \exp(\mathbf{a}_{it}^\top \boldsymbol{\theta} + b_{it})\right)$ where $\mathbf{a}_{it}$, $b_{it}$ are chosen randomly from normal distributions for $t = 1, 2$ and uniformly for $t = 3$. We approximate the Gibbs distributions $\exp(-\ell_t(\boldsymbol{\theta}))$ using the mixture-of-Gaussian algorithm described in Lin et al. (2019, Section 4.1) with full Hessians, and we use the EM algorithm described in Eq. 14 to find the mode of the mixture.

### B.2  MERGING LOGISTIC REGRESSION MODELS

For Simple Merging we train each model using gradient-descent with learning-rate $\rho = 3.0$ for 2500 iterations, and use $\sum_t \alpha_t \boldsymbol{\theta}_t$ to obtain each $\hat{\boldsymbol{\theta}}_{\boldsymbol{\alpha}}$. For the full-Gaussian method, which we use for Hessian-Weighted merging, we implement the variational online Newton method described in Khan & Rue (2023, Section 1.3.2). We set the learning-rate $\rho_t = 0.1$, perform 3 Monte-Carlo samples to estimate the expected gradient and Hessian and run for 25 iterations. The parameters of the merged model are obtained via Eq. 12. The mixture of full-Gaussian trains each model by the method described in Lin et al. (2019, Section 4.1) with a 20 component mixture. We set the algorithm's learning-rate of $\beta = 0.02$ for the mean $\boldsymbol{\mu}$ and precision $\boldsymbol{\Sigma}^{-1}$, $\beta = 3 \times 10^{-6}$ for the mixture weights $\pi$, while Monte-Carlo samples number of iterations are the same as full-Gaussian. The test-accuracy in Fig. 7, Fig. 8 and Fig. 9 is plotted on a grid $\boldsymbol{\Delta}$ with uniform spacing 0.02. The tasks are for imbalanced (MNIST Imb.) (T1: $\{0, 1\}$, T2: $\{2, 3, 4\}$ and T3: $\{5, 6, 7, 8, 9\}$); and for balanced (MNIST Bal.): (T1: $\{0, 1, 2\}$, T2: $\{3, 4, 5, 6\}$, T3: $\{7, 8, 9\}\}$).

For training the multitask models, perform finetuning we used nodes on a DGX-1 cluster with Nvidia V100 cards with 16GB of VRAM

### B.3  MERGING VISION MODELS ON CIFAR-10

We pretrain the ResNet-20 model by running the IVON optimizer for 1000 epochs with 5 Monte-Carlo samples to estimate expected gradients and Hessians and use IVON-HESS for estimates of the Pareto front. The hyperparameters of IVON are set as follows: learning-rate $\alpha = 0.1$, momentum $\beta = (0.9, 0.9999)$, weight-decay $\delta = 10^{-3}$ and temperature/sample-size weighting $\lambda = 50000$. The batch-size is set to 50 and the estimated Hessian is initialized to 0.1.

The individual models are also finetuned with IVON, initialized at the pretrained posterior for $\{25, 50, 75, 100, 125, 150\}$ steps over 5 random seeds to obtain a soup of 30 models for each task. This is similar to a mix of SoupIVON-Hess and MultiIVON-Hess. Each step processes 1000 examples, where the batch-size is set to 50. No weight-decay is used for finetuning, and we use a smaller learning-rate $\alpha = 0.01$. For the batch solution, we finetune on all data for 250 steps with the same hyperparameters.

The merged models $\hat{\boldsymbol{\theta}}_{\boldsymbol{\alpha}}$ are computed using Alg. 1, where we take the models across 5 random seeds with $\{100, 150\}$, $\{75, 100, 125, 150\}$ and $\{25, 50, 75, 100, 125, 150\}$ steps for the mixtures of size 10, 20 and 30. The test-accuracy in Fig. 2 is plotted on a grid $\boldsymbol{\Delta}$ with uniform spacing 0.1.

For training the multitask models, perform finetuning we used nodes on a DGX-2 cluster with Nvidia V100 cards with 32GB of VRAM.

### B.4  MERGING VISION TRANSFORMERS

The pretrained and finetuned checkpoints of ViT-B/32 a model based on CLIP (Radford et al., 2021) on these downstream tasks (GTSRB (Houben et al., 2013), RESISC45 (Cheng et al., 2017), SVHN (Netzer et al., 2011), EuroSAT (Helber et al., 2019), Stanford Cars (Krause et al., 2013) and SUN397 (Xiao et al., 2010)) were obtained based on the code from `https://github.com/mlfoundations/task_vectors`. The squared-gradients approximation for the Hessian-Weighted merge with ADAMW-SG is computed by $\sum_i \nabla \ell_i(\boldsymbol{\theta}_t)^2$, where $i$ is a sum over data examples from the training data.

To generate the exact solution contours we start from the pretrained checkpoint and finetune on the joint datasets with weights obtained by sampling from a grid $\mathbf{\Delta}$ with spacing 0.05. The optimizer is AdamW, with learning rate of $10^{-5}$, set $(\beta_1, \beta_2) = (0.9, 0.999)$ and decay the learning rate to 0 using a cosine decay with 500 warmup steps. Training is done for 15 epochs on GTSRB, RESISC45 and SVHN, while for EuroSAT, Stanford Cars and SUN397 this was set to 35, in both experiments batch size is 128.

For training the multitask models, perform finetuning we used nodes on a DGX-2 cluster with Nvidia V100 cards with 32GB of VRAM. To evaluate the merged models we employed nodes on a DGX-1 cluster with Nvidia V100 cards with 16GB of VRAM.

## B.5 MERGING MASKED LANGUAGE MODELS

We pretrain RoBERTa with 125M parameters using AdamW on the IMDB dataset for sentiment classification (Maas et al., 2011). We use a learning rate of $10^{-5}$ and set $(\beta_1, \beta_2) = (0.9, 0.999)$ and decay the learning rate to 0 using a cosine decay with 100 warmup steps. Training is done for 2 epochs with a batch size of 16.

We then finetune this model on Rotten Tomatoes (Pang & Lee, 2005), SST-2 (Socher et al., 2013), and Yelp (Zhang et al., 2015), and train with a learning rate of $5 \cdot 10^{-6}$ using a batch size of 16 and for 5, 5, and 2 epochs each. We subsample the data of Yelp by taking the first $20\%$ of the training data to ease computational burden.

We do not use any weight decay in pretraining or finetuning. The squared gradient approximation is calculated by doing one pass over the training data of each model and squaring the single-example gradients for ADAMW-SG.

For the batch solution, we finetune for 3 epochs on the concatenation of the above-mentioned training data after pretraining on IMDB as described above. Again, we use a learning rate of $5 \cdot 10^{-6}$ and a batch size of 16. Evaluation is done by averaging the accuracies over each individual dataset to weigh each dataset the same.

The simplex in Fig. 5a is obtained by sampling from a grid $\mathbf{\Delta}$ with spacing 0.05. For the joint solution we use a spacing of 0.1 due to the heavy computational load. The simple merged models are obtained using Eq. 9. For diagonal Gaussians, we use the Hessian-based weighting of Daheim et al. (2024). For merging with mixtures of Gaussians we use $K = 3$ and simply train using a different random seed to change the data order. All finetunings are done on a single NVIDIA GPU with as little as 12GB GPU memory and can be performed in under 2h.

## B.6 MERGING LLMS FOR MACHINE TRANSLATION

We finetune GEMMA-2B-it (Gemma Team, 2024b) on the IWSLT2017 de-en and fr-en splits (Cettolo et al., 2017). Due to the model size we use LoRA (Hu et al., 2022) to finetune the models which amounts to ca. 0.9M of new trainable parameters. The rest of the network is kept frozen. Accordingly, only the LoRA weights are merged and the base model untouched.

We train the models using IVON with a learning rate of 0.05, $(\beta_1, \beta_2) = (0.9, 0.99995)$, an effective sample size of $1 \cdot 10^7$ for the single-task and $2 \cdot 10^7$ for the multitask model. We clip gradients element-wise to $1 \cdot 10^3$ and to unit norm and use a weight decay of $10^{-6}$.

For the Hessian-weighted merging we use IVON-HESS. A comparison to using squared gradients instead is found in App. C.3. In all experiments, we use a grid with equal spacing of $\alpha_1 \in [0.0, 0.05, \ldots, 1.0]$ and set $\alpha_2 = 1.0 - \alpha_1$. For mixture of Gaussians we again use $K = 3$ and alternate the random seed. For generation, we use greedy decoding from the model.

All trainings are performed on a single NVIDIA GPU with up to 80GB GPU memory.

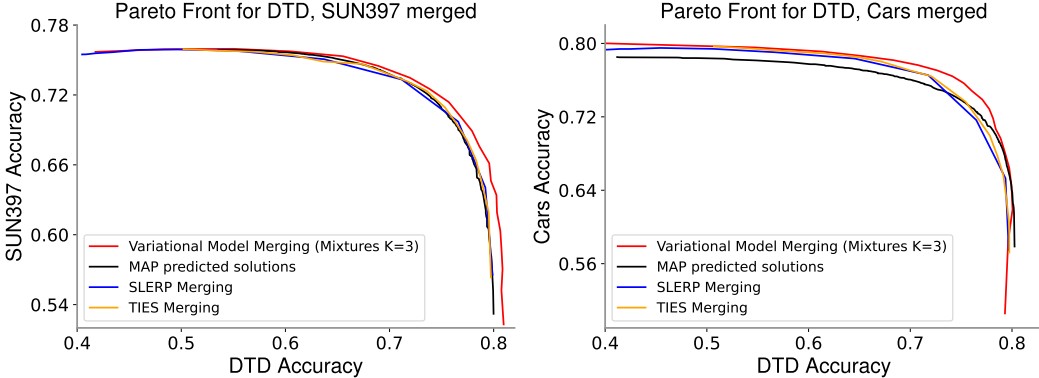

Figure 6: Pareto fronts obtained with a three component Mixture-Weighted Merging, compared to MAP with Task-Arithmetic, TIES and SLERP. Around the knee of the Pareto front, we see our proposed method obtains better accuracy on both DTD-SUN397 and DTD-Stanford Cars compared to the other methods.

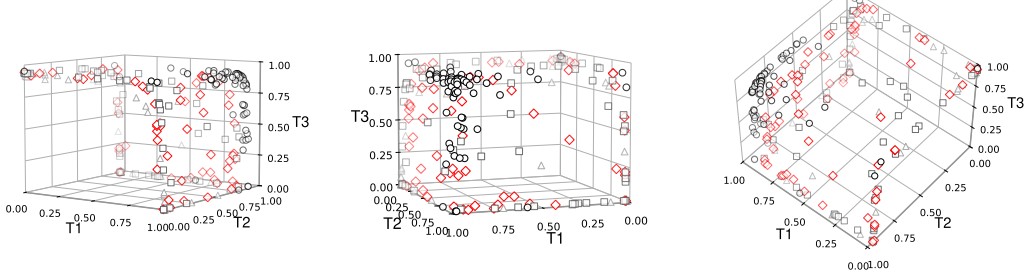

Figure 7: Pareto fronts using a logistic regression model for training on an balanced split of MNIST. We compare the Pareto fronts of Multitask finetuning (○), Mixture of Gaussian merging (◇), Hessian-Weighted Merging (□) and Simple merging (△).

## C  ADDITIONAL RESULTS

### C.1  COMPARISON WITH OTHER MERGING METHODS

We compare our variational model merging against some other baselines like TIES-merging (Yadav et al., 2023) and SLERP (Shoemake, 1985). We also include MAP (Model Merging with Amortized Pareto Front) (Li et al., 2025a) paired with TA, and use the settings from Section 4.3 in that work. We see in Fig. 6 a better performance by our Mixture-weighted merging ($K = 3$), indicating that a better posterior captures the global shape of the loss landscape and helps producing a better approximation of the front in regions far from the extremes, i.e performance of the models used in the merge.

### C.2  MULTITASK LEARNING ON MNIST

We consider MNIST broken into three tasks, each consisting of a different and disjoint subset of classes. We use a logistic regression model in two settings: one imbalanced set where number of classes per tasks vary and a balanced set. In both cases we compare isotropic Gaussian (Simple Merging), full Gaussian (Hessian-Weighted), and mixture-of-Gaussian (Mixture-Weighted) posteriors. To compute the posterior approximations and surrogate functions, we use VON (Khan & Rue, 2023) and for mixture-of-Gaussians the joint learning algorithm from Lin et al. (2019), both with full Hessian. For Hessian-Weighted merging we use Eq. 12 for all $\alpha$, for the mixture-of-Gaussians we use the EM algorithm outlined in Eq. 14.

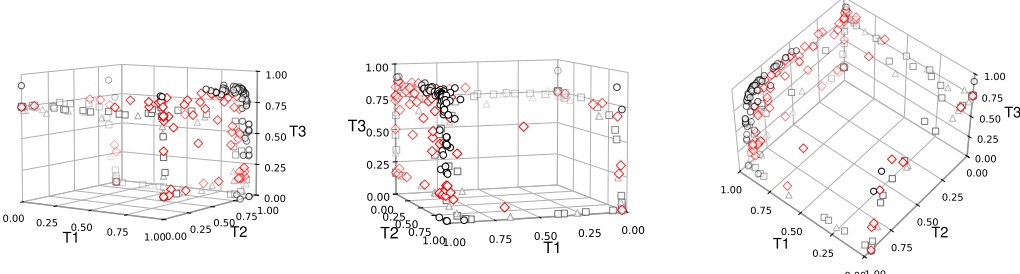

Figure 8: Pareto fronts using a logistic regression model for training on an imbalanced split of MNIST. We compare the Pareto fronts of Multitask finetuning (○), Mixture of Gaussian merging (◇), Hessian-Weighted Merging (□) and Simple merging (△).

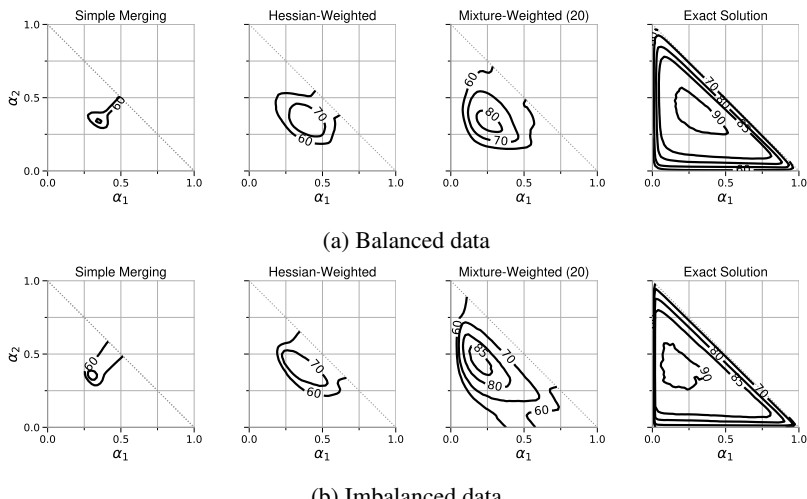

(a) Balanced data

(b) Imbalanced data

Figure 9: Results showing the full shape of trade-offs when using a logistic regression model for training on an imbalanced split of MNIST. We compare the Pareto fronts of Multitask finetuning, Mixture of Gaussian merging, Hessian-Weighted Merging and Simple merging.

While Fig. 7 and Fig. 8 show a comparison of Pareto fronts, Fig. 9 shows the full trade-off also for points that are dominated by others from the same method and are off the Pareto front. In both cases we find that more expressive posteriors yield better results. Especially when looking at the full shape one notices quickly that better posteriors cover broader regions with good performance. For the balanced setting there is no skew and the better combinations seem to concentrate around the center of the simplex, which we see in Figure Fig. 9a is captured by all methods, however the more complex posterior approximation allows Hessian-Weighting and Mixture-Weighting to show that multiple combinations even beyond the center are also interesting which Simple Merging fails to convey.

## C.3    COMPARISON OF HESSIAN APPROXIMATIONS FOR LLM MERGING

Fig. 10 shows a comparison of using the squared gradient approximation of the (diagonal) Fisher and the diagonal Hessian approximation obtained with IVON for Hessian-Weighted merging. Both methods are comparable and provide good previews for multitask finetuning. However, the Hessian approximation from IVON comes for free during training while the squared gradient approximation incurs overhead due to requiring an additional forward pass over at least a subset of the training data after training.

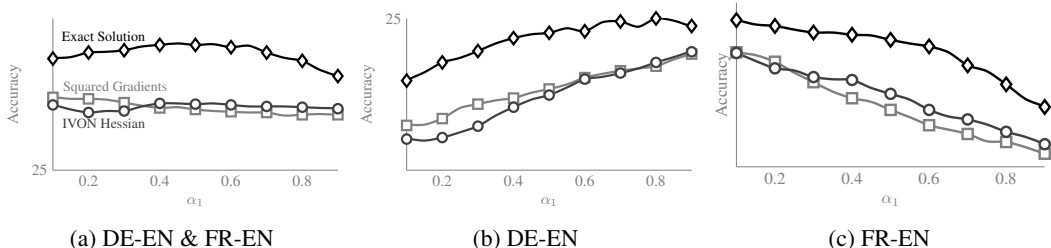

(a) DE-EN & FR-EN        (b) DE-EN        (c) FR-EN

Figure 10: Here, we merge LoRA-finetuned GEMMA-2B models trained on IWSLT2017de-en and IWSLT2017fr-en. We show a comparison of using the squared gradient approximation of the (diagonal) Fisher and the diagonal Hessian approximation obtained with IVON for Hessian-weighted merging. Both methods perform similarly and could be used effectively for estimating Pareto fronts.

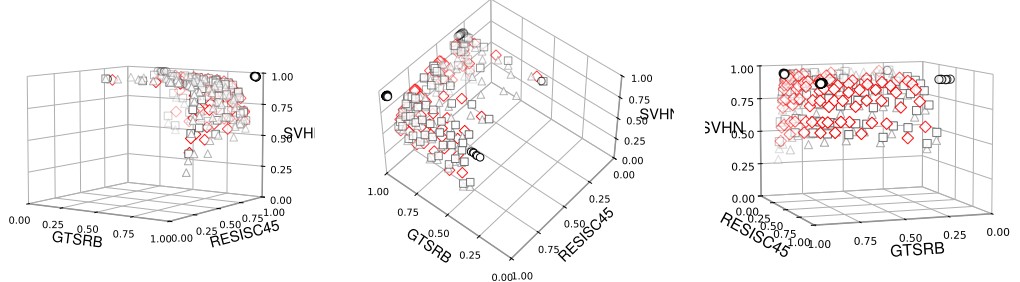

Figure 11: Results using ViT-B/32 on GTSRB, RESISC45, SVHN. Again, multitask finetuning (○) results in three clusters. Mixture of Gaussian merging (◇) appears less uniform than Hessian-Weighted Merging (□) and Simple merging (△) and is closing in on the clusters found by multitask training, potentially due to the more flexible posterior.

## C.4 COMPARISON OF ViT MERGING ON GTSRB, RESISC45, SVHN

Fig. 11 shows a comparison of ViT's trained on GTSRB, RESISC45, SVHN, as described in Sec. 4.2. We refer to a discussion of results there, since the conclusions and setting are the same.

## C.5 WEIGHTS FOUND VIA PARETO FRONT ESTIMATES

Table 2 summarizes the $\alpha$ combinations that achieve the maximum accuracy according to each method used to estimate Pareto fronts. From the results in Table 1, we see that estimates via merging obtain weights that achieve a comparably high accuracy, even if they do not always match the ones found by several runs of scalarization and multitask finetuning. The accuracy also improves with better posterior approximations. Note that the cost to prepare the individual models per task to do previews is equivalent to one run of multitask finetuning. Afterwards, several weights can be tried without any extra training and at negligible overhead using merging. Exploring such weights through multitask finetuning becomes prohibitively expensive with larger models and larger amounts of tasks.

## C.6 IMPROVEMENT IN PER TASK ACCURACY

We showcase per task accuracy improvement obtained by more flexible posteriors for many combinations of $\alpha$ in Figs. 12 to 15. Mixture-Weighted merging is compared against Hessian-Weighted (top) and Simple Merging (bottom) per task. Taking the difference in achieved test accuracy between methods, we show inside a simplex red contours when Mixture performs better, grey when there is no difference, and blue when the other method obtained a higher accuracy.

In general, we see that while mixtures show improvement over the other merging methods, is not for all tasks and for all $\alpha$ combinations. However, they uncover models that the other methods were not able to find.

Table 2: We report the best $\alpha$ combination found through Pareto front estimates for each experiment. As seen in Table 1, for most experiments, even if the $\alpha$ weights do not exactly match the multitask finetuned ones, accuracy is still high. Note that generating Pareto front estimates for several $\alpha$ using merging takes seconds and can guide the practitioner towards high accuracy weightings, while a full sweep through many combinations for multitask finetuning becomes prohibitive with larger models. Datasets[1]: GTSRB, RESISC45, SVHN; Datasets[2]: EuroSAT, Cars, SUN397.

| Model | Logistic | | | | ResNet-20 | | ViT-B/32 | | ViT-B/32 | | RoBERTa | | GEMMA-2B |
|---|---|---|---|---|---|---|---|---|---|---|---|---|---|
| Tasks | MNIST Imb. | | MNIST Bal. | | CIFAR-10 | | Datasets[1] | | Datasets[2] | | RT, SST-2, Yelp | | IWSLT2017 |
| $\alpha$ | $\alpha_1$ | $\alpha_2$ | $\alpha_1$ | $\alpha_2$ | $\alpha_1$ | $\alpha_2$ | $\alpha_1$ | $\alpha_2$ | $\alpha_1$ | $\alpha_2$ | $\alpha_1$ | $\alpha_2$ | $\alpha_1$ |
| Simple Merging | 0.34 | 0.34 | 0.34 | 0.34 | 0.40 | 0.20 | 0.35 | 0.35 | 0.35 | 0.40 | 0.10 | 0.55 | 0.05 |
| Hessian Weighted | 0.42 | 0.36 | 0.40 | 0.34 | 0.30 | 0.40 | 0.35 | 0.35 | 0.35 | 0.45 | 0.15 | 0.50 | 0.40 |
| Mixture Weighted | 0.14 | 0.46 | 0.24 | 0.34 | 0.40 | 0.20 | 0.25 | 0.3 | 0.15 | 0.05 | 0.05 | 0.40 | 0.05 |
| Multitask Finetuning | 0.12 | 0.46 | 0.28 | 0.42 | 0.60 | 0.10 | 0.15 | 0.70 | 0.10 | 0.75 | 0.15 | 0.40 | 0.45 |

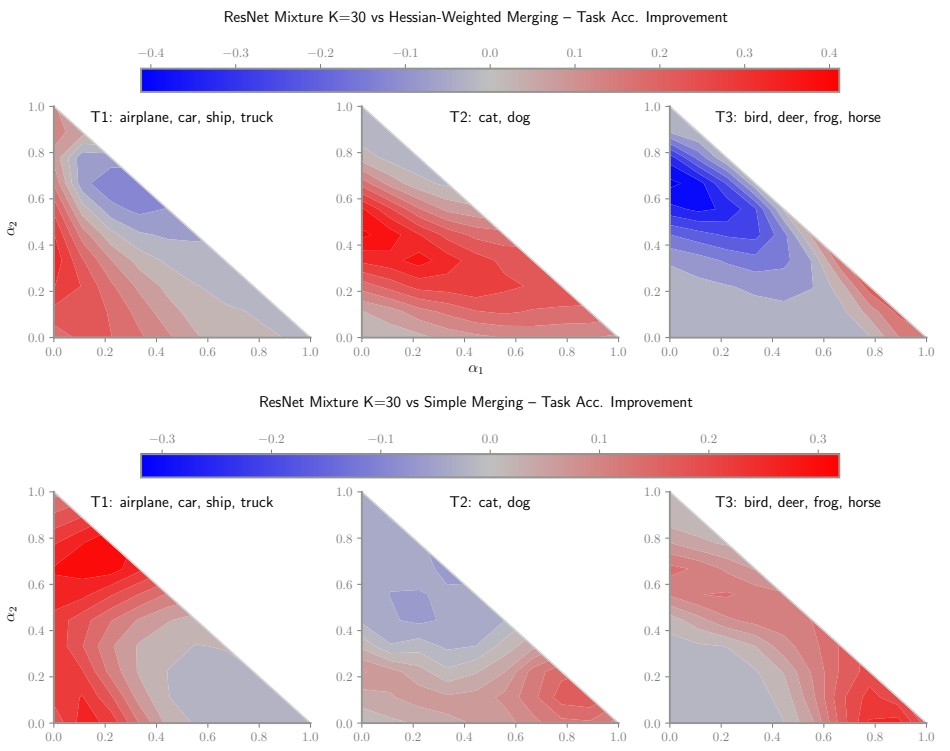

Figure 12: Test accuracy improvement for each $\alpha$ for ResNet-20 on split CIFAR10. Red shows gained accuracy by using mixtures, blue when mixtures were not better.

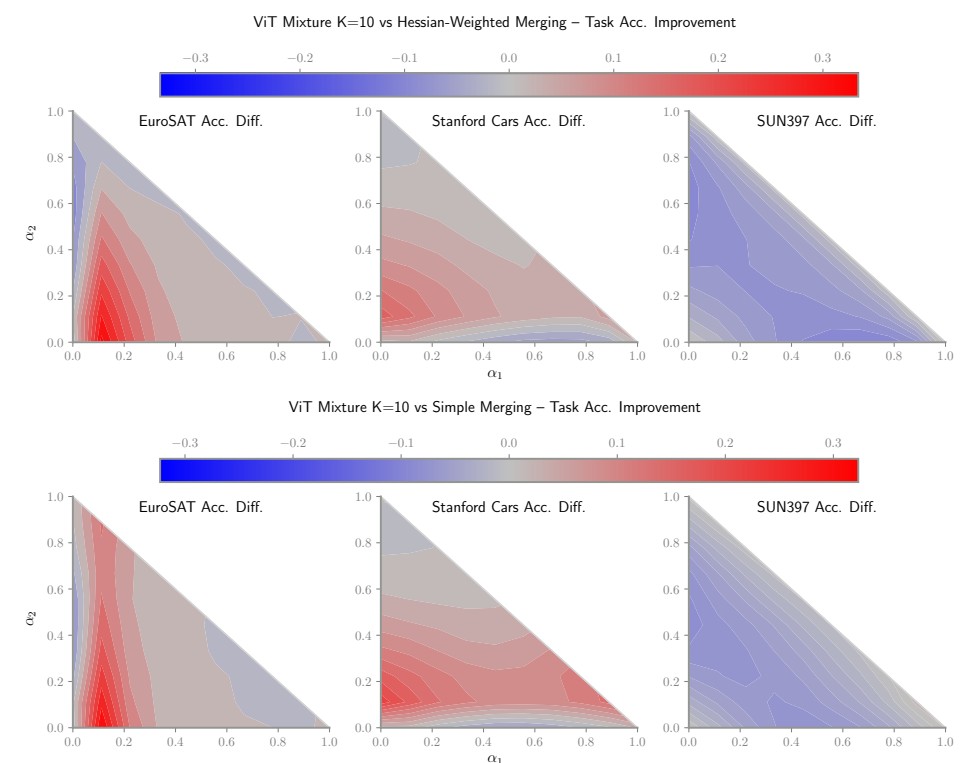

Figure 13: Test accuracy improvement for each $\alpha$ for ViT-B/32 on Euro-Cars-SUN397. Red shows gained accuracy by using mixtures, blue when mixtures were not better.

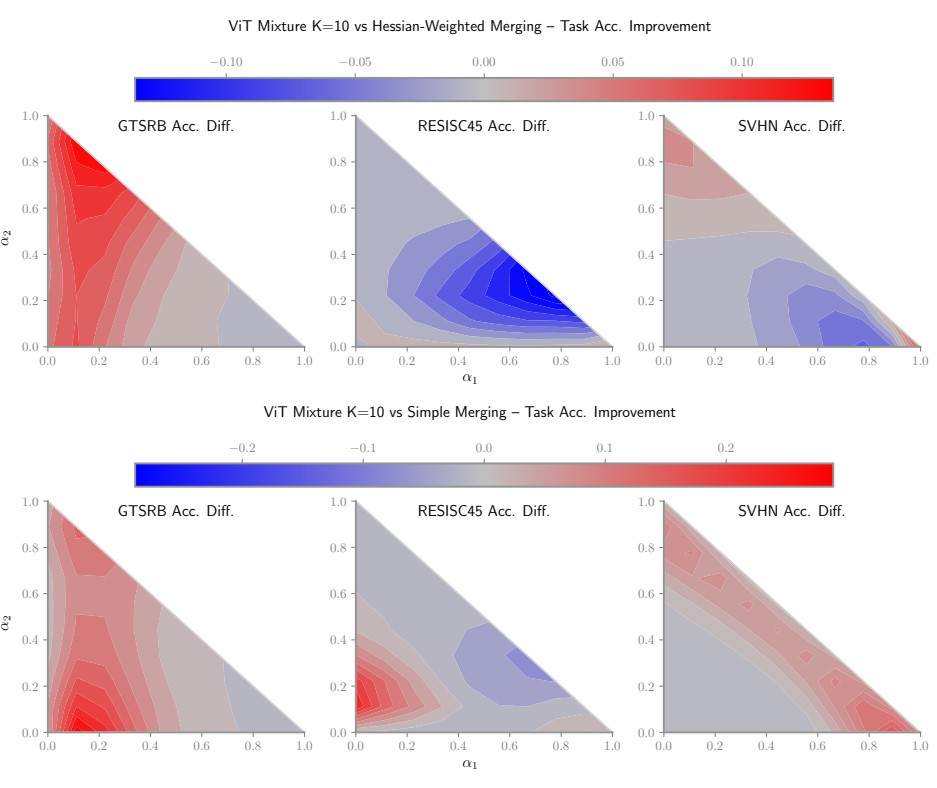

Figure 14: Test accuracy improvement for each $\alpha$ for ViT-B/32 on GTSRB-RESISC45-SVHN. Red shows gained accuracy by using mixtures, blue when mixtures were not better.

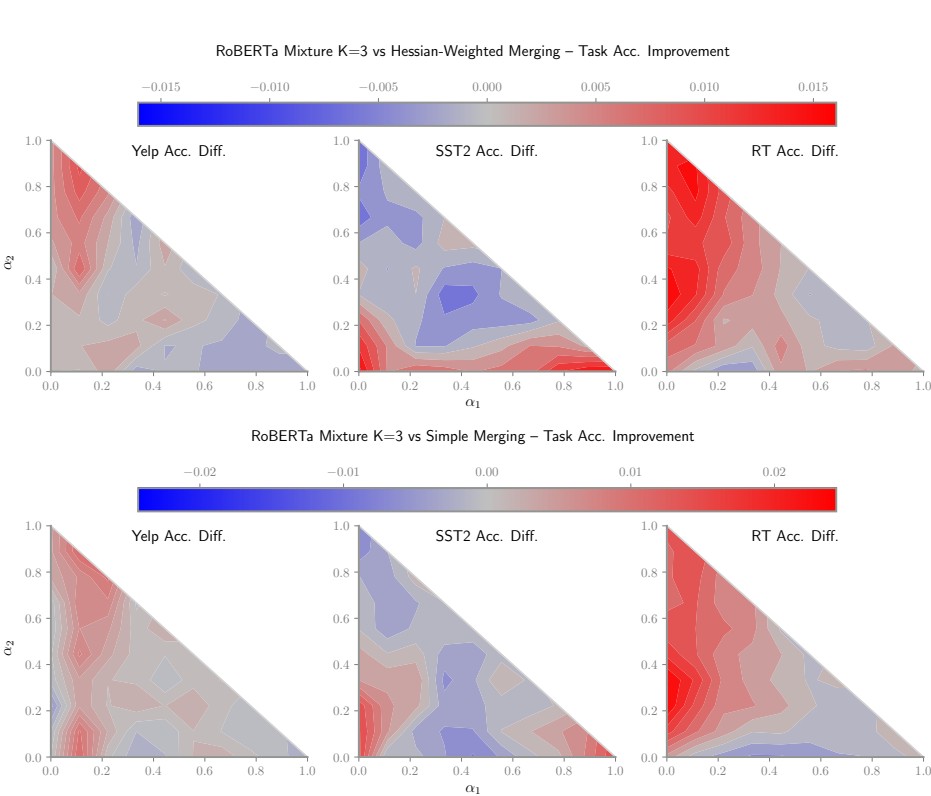

Figure 15: Test accuracy improvement for each $\alpha$ for RoBERTa on Yelp-SST2-RottenTomatoes. Red shows gained accuracy by using mixtures, blue when mixtures were not better.

