# OpenReview forum: "Variational Model Merging for Pareto Front Estimation in Multitask Finetuning"
_ICLR.cc/2026/Conference — Submitted to ICLR 2026_

### Official Review · Reviewer_ChLM · 2025-11-01

**Soundness:** 3
**Presentation:** 3
**Contribution:** 2
**Rating:** 4
**Confidence:** 4

**Summary:**

The paper frames *model merging* as approximate Bayesian inference to cheaply preview the Pareto set for multitask finetuning. Starting from task-specific posteriors, it proposes estimating Pareto solutions by maximizing a merged posterior. This unifies prior weight-averaging (simple averaging / task arithmetic) with Hessian-weighted schemes and introduces a mixture-of-Gaussians variant solved via a lightweight EM procedure. Empirically, across CIFAR-10 (ResNet-20), CLIP ViT-B/32 transfers, RoBERTa sentiment, and GEMMA-2B LoRA MT, more expressive posteriors (diagonal Fisher or Hessian to mixtures) produce Pareto fronts closer to multitask finetuning while being much cheaper than retraining across many values.

**Strengths:**

The paper clearly derives how scalarized multi-objective training corresponds to MAP under a merged posterior; it shows that common merging tricks are special cases of exponential-family surrogates (e.g., simple averaging from isotropic Gaussian; Hessian-weighted from full Gaussian). This gives a principled recipe rather than ad-hoc formulas. Consistent empirical trend: Across tasks and model families, more flexible posterior means better front.

**Weaknesses:**

- Approximation stack is heavy and sometimes crude. Many experiments rely on diagonal Hessians/Fishers or squared-gradient proxies, which can mischaracterize curvature and interactions (acknowledged by the authors).
- Cost shifts rather than disappears. MoG requires K runs per task and a few EM steps. While still cheaper than dense alpha sweeps, for large T and K this becomes significant; the paper reports K=3–30, which is nontrivial in big models.

**Questions:**

- Hessian quality vs. downstream accuracy. The argument that IVON supplies a “free” diagonal Hessian is practical, but no controlled study links Hessian quality to Pareto-front error beyond rough accuracy differences. A calibration plot (front error vs. curvature error) would strengthen the causal story. (Setup & comparisons.)
- The method assumes task-specific posteriors are compatible under a common prior. In settings with strong parameter non-identifiability or sharp mode shifts (e.g., safety vs. creativity in LLMs), merging may land off-manifold; the paper hints at such gaps (e.g., shape mismatches) but doesn’t delineate failure modes or detection heuristics.
- Reported times exclude the up-front cost of training each task model (and K variants for mixtures). For large T, the one-time cost may approach/ exceed a modest grid of multitask runs; a more apples-to-apples wall-clock accounting would help.

---

> ### Author Response · Authors · 2025-11-21
>
> We appreciate the time taken by the reviewer to provide their feedback and thank the reviewer for emphasizing that our work “gives a principled recipe rather than ad-hoc formulas” with a “consistent empirical trend”. We address their concerns below and hope the reviewer can reconsider their valuation of our work.
>
> > “Approximation stack is heavy and sometimes crude. Many experiments rely on diagonal Hessians/Fishers or squared-gradient proxies, which can mischaracterize curvature and interactions (acknowledged by the authors).”
>
> We use diagonal approximations because there are currently no variational methods for non-diagonal covariances which can consistently achieve good results at large scale.
> For example, [1] shows that noisy K-FAC [2] is sometimes drastically outperformed by diagonal covariance. However, we are optimistic that merging methods derived from our framework can benefit from non-diagonal covariance optimizers being made more practical in the future, and our framework indicates that better performance is to be expected.
>
> > “Cost shifts rather than disappears. MoG requires K runs per task and a few EM steps. While still cheaper than dense alpha sweeps, for large T and K this becomes significant; the paper reports K=3–30, which is nontrivial in big models.”
>
> We note that the cost is still significantly reduced compared to retraining which is the goal of our paper. For $D$ grid points, retraining has an exponential complexity of $\mathcal{O}(D^T)$ while mixtures have at most polynomial complexity of $\mathcal{O}(K\cdot T)$. MoGs with $K$ components can also be constructed with fewer than $K$ training runs. In Fig. 2 we show a result, where mixture components are obtained by reusing checkpoints over the training trajectory. Such a strategy can reduce the cost during training to as far as $\mathcal{O}(T)$ if checkpoints of only one training run are used. We have emphasized this more in our work, thank you for raising this point!
>
> >Q1: “Hessian quality vs. downstream accuracy. The argument that IVON supplies a “free” diagonal Hessian is practical, but no controlled study links Hessian quality to Pareto-front error beyond rough accuracy differences. A calibration plot (front error vs. curvature error) would strengthen the causal story. (Setup & comparisons.)”
>
> It is unfortunately very difficult to precisely control the quality of Hessian approximations in the variational bound, because the only variational methods that work at a large scale use diagonal Hessian. Therefore, providing such a study is not straightforward but from a theoretical point of view it is clear that better Hessian approximations improve the variational approximation and should therefore in general lead to better downstream performance.
>
> >Q2: The method assumes task-specific posteriors are compatible under a common prior. In settings with strong parameter non-identifiability or sharp mode shifts (e.g., safety vs. creativity in LLMs), merging may land off-manifold; the paper hints at such gaps (e.g., shape mismatches) but doesn’t delineate failure modes or detection heuristics.
>
> The method itself does not require a common prior. In Eq. 3 each posterior can be trained under a different prior, it would suffice to divide the posterior of task $t$ by the used prior ( $p_t(\theta) / p_{0,t}(\theta)$) and then add a new common $p_0(\theta)$.
> Any model merging method may land off-manifold and there is unfortunately no direct way of preventing this. This is also due to the approximation error made in model merging which can be large, for example, if there is interference between datasets. We have added a discussion of this in Sec. 3.4. We agree that understanding such failure modes further is an important issue but this requires a study of its own and is out of scope for this work.
>
> >Q3: “Reported times exclude the up-front cost of training each task model (and K variants for mixtures). For large T, the one-time cost may approach/ exceed a modest grid of multitask runs; a more apples-to-apples wall-clock accounting would help.”
>
> Our comparison is an apples-to-apples comparison. We also report training times, for example at the end of Secs. 4.2 and 4.4. However, exact training times are arguably less important than overall complexity, as a standard multitask training run always takes just as much time as training on each dataset independently. These costs are discussed in Sec. 3.5.
>
> ## References
>
> [1] Osawa et al., “Practical Deep Learning with Bayesian Principles”, NeurIPS 2019
>
> [2] Zhang et al., “Noisy natural gradient as variational inference”, ICML 2018

---

> > ### Author Response · Authors · 2025-11-27
> > **Gentle Reminder.**
> >
> > Dear Reviewer,
> >
> > Thank you again for your review.
> >
> >
> > Since the discussion period will be closing soon, please let us know if there are any further questions.

---

### Official Review · Reviewer_J8tu · 2025-11-01

**Soundness:** 3
**Presentation:** 3
**Contribution:** 3
**Rating:** 6
**Confidence:** 3

**Summary:**

This paper proposes variational model merging, a Bayesian approach to estimate Pareto fronts in multitask finetuning by merging task-specific posterior approximations. The key insight is that more flexible posterior families (e.g., full Gaussians, mixture of Gaussians) yield better Pareto front estimates than simpler ones

**Strengths:**

1. Novel theoretical framework: Connecting model merging to Bayesian posterior fusion is novel and provides a principled way to derive new merging strategies. The variational perspective naturally explains why different merging methods exist and how to improve them.
2. Clear theoretical contribution: Theorem showing that more flexible posteriors necessarily yield better estimates is valuable, with the error reduction property being particularly insightful.
3. Comprehensive experiments: Testing on diverse architectures and tasks (vision, NLP, translation) demonstrates broad applicability.

**Weaknesses:**

1. Missing bounds on approximation quality relative to true Pareto front. Also, can authors provide a formal connection between posterior quality and Pareto front accuracy? In other words, a bound on how the approximation quality translates to Pareto solution quality. Currently, the paper only shows empirically that better posteriors help, but doesn't prove: how much they help and when they're guaranteed to help?
2. computational costs.  1. mixture methods require K times more models, which is expensive for large models. 2. As author already state, Hessian approximation is a bottleneck for large model merging, even diagonal approximations require O(P) storage. Analysis of computational costs for various

**Questions:**

Can this framework handle constraints or preferences on the Pareto front?

---

> ### Author Response · Authors · 2025-11-21
>
> We thank the reviewer for their effort and for pointing out the novel framework, clear theoretical contribution and comprehensive experiments. We address their concerns below.
>
> > “Missing bounds on approximation quality relative to true Pareto front. Also, can authors provide a formal connection between posterior quality and Pareto front accuracy? In other words, a bound on how the approximation quality translates to Pareto solution quality. Currently, the paper only shows empirically that better posteriors help, but doesn't prove: how much they help and when they're guaranteed to help?”
>
> We have added a new Sec. 3.4 which provides a theoretical connection between the expressiveness of the posterior and the accuracy of the Pareto front. Essentially, a more expressive posterior already includes less expressive posteriors as a subset (for example, diagonal Gaussians can always be expressed as full Gaussians) and necessarily lead to a better posterior approximation and better surrogates. For example, isotropic Gaussians lead to linear surrogates but full Gaussians lead to quadratic surrogates. This holds for each possible $\alpha$ which implies a Pareto front estimation that is at least as good. We refer the reviewer to Sec 3.4 for more details and are happy to discuss this further.
>
> > “computational costs. 1. mixture methods require K times more models, which is expensive for large models. 2. As author already state, Hessian approximation is a bottleneck for large model merging, even diagonal approximations require O(P) storage. Analysis of computational costs for various”
>
> These are not bottlenecks. Both mixtures and Hessian approximations are cheap.
> First, while mixtures need more models, these can be obtained within a single or just a few training runs by taking checkpoints. We discuss this strategy in Sec. 3.5. and show results with a similar approach in Fig. 2. Second, Hessian approximations are obtained for free, for example, through AdamW or IVON’s scale vector.
>
>
> > “Can this framework handle constraints or preferences on the Pareto front?”
>
> Yes, constraints or preferences could be handled. For example, one can discard solutions that violate constraints or restrict the search space to $\alpha$s that, for example, match user preferences.
>
> We again thank the reviewer for their review and kindly request them to reconsider their evaluation should this have addressed their concerns.

---

> > ### Author Response · Authors · 2025-11-27
> > **Gentle Reminder.**
> >
> > Dear Reviewer,
> >
> > Thank you again for your review.
> >
> >
> > Since the discussion period will be closing soon, please let us know if there are any further questions.

---

> > > ### Comment · Reviewer_J8tu · 2025-11-28
> > >
> > > Thank authors for the response. I will maintain my score. And AdamW uses second-moment information of the gradients, but it is not a Hessian (or inverse-Hessian) approximation in the same sense as Newton or quasi-Newton methods.

---

### Official Review · Reviewer_nLqB · 2025-11-01

**Soundness:** 3
**Presentation:** 3
**Contribution:** 3
**Rating:** 4
**Confidence:** 4

**Summary:**

The authors employed a Bayesian model-merging approach that efficiently explores various weighting configurations without requiring full retraining for each one. Their method relies on two key components: model merging, which combines the parameters of models individually trained on separate tasks instead of retraining for every configuration, and a Bayesian framework, which enhances the merging process by developing improved surrogate functions for the multitask learning objective. This allows practitioners to effectively explore task-weighting options and find high-performing models at a fraction of the computational cost of traditional retraining.

**Strengths:**

- The paper's primary strength is its novel conceptualization of model merging as a variational Bayesian inference problem. This original framework is significant because it replaces ad-hoc merging heuristics with a foundation that both explains the relative performance of existing methods and provides a clear recipe for systematically designing new, more accurate ones.
- Extensive empirical validation on modern, large-scale architectures, including Vision Transformers and the GEMMA-2B LLM.

**Weaknesses:**

- The paper's primary goal is to provide "fast and cheap methods" to estimate the Pareto set. However, its best-performing and most novel method, Mixture-Weighted Merging (MultiIVON-Hess), has a training cost that scales linearly with the number of mixture components ($K$). This requires $K$ full training runs for each task, which creates a significant tension with the "cheap" objective.

**Questions:**

- The number of components $K$ seems to be a critical hyperparameter. The paper uses $K=30$ for ResNet, $K=10$ for ViT, and $K=3$ for RoBERTa and GEMMA. How was $K$ chosen for each experiment? Is there a principled way to select $K$?

---

> ### Author Response · Authors · 2025-11-21
>
> We thank the reviewer for their review and for pointing out the “novel conceptualization of model merging as a variational Bayesian problem” and the “[e]xtensive empirical validation on modern, large-scale architectures”. We address their concerns and questions regarding the hyperparameter $K$ and Mixture-Weighted merging method below and kindly request the reviewer to revise their score.
>
> > “The paper's primary goal is to provide "fast and cheap methods" to estimate the Pareto set. However, its best-performing and most novel method, Mixture-Weighted Merging (MultiIVON-Hess), has a training cost that scales linearly with the number of mixture components $(K)$. This requires $K$ full training runs for each task, which creates a significant tension with the "cheap" objective”
>
> Here are some clarifications that may help. ‘Cheap’ refers to the fact that we reduce the cost of exhaustive retraining (which is exponential) to a cost which is linear in the number of tasks. Specifically, the original cost of computing the Pareto set with $D$ grid points is exponential in the number of tasks ($D^T$) but we reduce this exponential complexity to a polynomial one ($T*K$). One mixture component $K=1$ is of course the cheapest, but increasing $K$ improves performance at only a linear cost. The training cost of $K$ can also be reduced even when using mixtures by simply taking checkpoints across the training trajectories, similar to model souping [1]. We use a similar strategy in Fig. 2 and have added it as a specific algorithm in Sec. 3.
>
> > “The number of components $K$ seems to be a critical hyperparameter. The paper uses $K=30$ for ResNet, $K=10$ for ViT, and $K=3$ for RoBERTa and GEMMA. How was $K$ chosen for each experiment? Is there a principled way to select $K$?”
>
> Yes, there are principled ways to do this, for example, we can use the ELBO values and optimize the marginal likelihood. In general, we expect larger $K$ to provide better results. Larger $K$ can also be achieved cheaply by taking checkpoints along the trajectory, as discussed above. In our paper, we determined $K$ heuristically based on the compute requirement of each problem.
>
> ## References
>
> [1] Wortsman et al., “Model soups: averaging weights of multiple fine-tuned models improves accuracy without increasing inference time”, ICML 2022

---

> > ### Author Response · Authors · 2025-11-27
> > **Gentle Reminder.**
> >
> > Dear Reviewer,
> >
> > Thank you again for your review.
> >
> >
> > Since the discussion period will be closing soon, please let us know if there are any further questions.

---

### Official Review · Reviewer_fbTv · 2025-11-02

**Soundness:** 3
**Presentation:** 3
**Contribution:** 3
**Rating:** 6
**Confidence:** 4

**Summary:**

This paper proposes a Bayesian method for model merging. The goal is to approach Pareto front of multitask by efficiently and approximately computing the posterior with mixture of Gaussian. As a consequence, this method balance the efficiency that is lacking by full Gaussian posterior and the utility that is lacking by isotropic Gaussian. Experiments on transformers have shown some improvement that model merging gets closer to multitask finetuning.

**Strengths:**

This paper is clearly written with good introduction and motivation. The Bayesian approach makes sense to me and is original as far as I can tell. Figure 1 really does a good job highlighting the idea. Section 3.2 positions this method appropriately in the literature. Overall the quality is good, with all the derivation and reasoning being sound.

**Weaknesses:**

1. Methodology

Section 3.4 lists three versions and different experiments seem to use different versions. It would be beneficial to converge to one method if possible for practitioners. If not, can the authors summarize the applicability of each version?

2. Weaker performance than multitask finetuning

The message from this work is two-fold: variational model merging is better than previous model merging, but it is still worse than multitask finetuning (see Table 1 and Figure 5b). While the second part is not a positive result, I think it is very valuable. However, to take the second conclusion seriously, the overall method between line 292-294 may need a 4th step: launch multitask finetuning for the Pareto estimates.

3. Computational cost

This method is still computationally heavy, e.g. finetuning T models for T tasks. While many model merging methods are costly, this painpoint is not yet alleviated by this method so I think the significance is not great.

**Questions:**

See weaknesses.

---

> ### Author Response · Authors · 2025-11-21
>
> We thank the reviewer for their review and especially for highlighting the clarity and soundness. Below we address their concerns.
>
> > “Section 3.4 lists three versions and different experiments seem to use different versions. It would be beneficial to converge to one method if possible for practitioners. If not, can the authors summarize the applicability of each version?”
>
> We do not believe that providing different versions is a weakness but rather a strength, because it allows the practitioner to choose the best version that is still within their compute and memory constraints but also provides good performance
> In general, we would expect more expressive methods (mixtures or Hessian-based merging) to work better than simpler ones (e.g. isotropic Gaussians) but mixtures may require more component models. We note though that the training cost for obtaining mixtures can be significantly reduced by taking checkpoints during training, as is now also discussed in Sec. 3.5.
>
> > “Weaker performance than multitask finetuning
> The message from this work is two-fold: variational model merging is better than previous model merging, but it is still worse than multitask finetuning (see Table 1 and Figure 5b). While the second part is not a positive result,  think it is very valuable. However, to take the second conclusion seriously, the overall method between line 292-294 may need a 4th step: launch multitask finetuning for the Pareto estimates.”
>
> Multitask fine tuning is the gold standard (and requires heavy retraining), so it is expected that any model merging method will be worse. We argue for this in the newly written Sec. 3.4 which describes why more flexible posteriors yield better approximations. We agree that Step 4 is necessary as described. In fact, we have mentioned this at the beginning of Sec. 3 and already provided results for this 4th step in gray in Table 1. To make this clearer, we have now also added it to the recipe explicitly, thank you for the suggestion!
>
> > “Computational cost
> This method is still computationally heavy, e.g. finetuning T models for T tasks. While many model merging methods are costly, this painpoint is not yet alleviated by this method so I think the significance is not great.”
>
> Our goal is not to cut down the costs of model merging but to provide methods for estimating Pareto fronts that are cheaper than retraining. Training $T$ models has the same cost as one multitask fine tuning run, but it then enables us to try out a large number of $\alpha$ combinations at no extra cost. Oftentimes, the total cost can even be smaller than $T$, for example, because training models exist on the web. We therefore do not agree with the criticism and believe it is true for any model merging work.
>
> We again thank the reviewer for their review! We would appreciate it if the reviewer could re-evaluate their score should their concerns be addressed.

---

> > ### Author Response · Authors · 2025-11-27
> > **Gentle Reminder.**
> >
> > Dear Reviewer,
> >
> > Thank you again for your review.
> >
> >
> > Since the discussion period will be closing soon, please let us know if there are any further questions.

---

### Author Response · Authors · 2025-12-04
**Message to AC**

Dear AC,

Thank you very much for taking over our submission.

Our work presents a *variational approach* for model merging that can be used to obtain arbitrarily accurate estimates of Pareto fronts by choosing flexible posterior structures.
We find this exciting because the connection between **variational Bayesian methods** and **Pareto front estimation** is not known in the literature but provides a clear recipe for designing accurate model merging strategies. This is of high practical relevance to many problems, for example, for estimating the scaling coefficients of data mixes in LLM training and all reviewers appreciate this new connection.

The primary concern seems to have been the compute requirements of our **novel mixture-based method**. In our responses (e.g. to **R-ChLM**), we have clarified that mixtures reduce the *exponential computational complexity* of retraining to an at most *polynomial* one and can be obtained cheaply via checkpointing at no overhead when compared to simpler strategies.
Among other smaller clarifications, we have also added a precise theoretical justification of why more flexible posteriors lead better approximations in Section 3.4, in response to **R-J8tu**.

We believe that we have addressed all concerns well and do not believe there to be any substantial remaining concerns but could unfortunately not get responses in time from most reviewers.
We would appreciate it if you could have a closer look when making a decision.

---

### Meta-Review · Area_Chair_jPiY · 2026-01-08

**Summary:**

This paper, proposing a variational Bayesian model merging method for Pareto front estimation in multitask finetuning, received four reviewers’ feedback focusing on three key concerns. First, computational costs emerged as a primary issue. Reviewers argued that the novel Mixture-Weighted Merging scales linearly with mixture components, requiring $K$ training runs per task—costly for large models. They also noted that upfront training of task-specific models might offset savings against exhaustive retraining. Second, theoretical and methodological gaps were highlighted. Reviewers pointed out the lack of formal bounds connecting posterior quality to Pareto front accuracy, as the paper only offers empirical evidence. They also questioned the use of multiple method versions without clear applicability guidelines, plus crude diagonal Hessian/Fisher approximations that may misrepresent model curvature. Third, practical limitations were raised. The method underperforms full multitask finetuning, and reviewers noted its failure to address edge cases like parameter non-identifiability in LLMs, where merging could yield off-manifold solutions with no detection heuristics. Although 2 of the reviewers give the score to 6, none reviewer shows strong support for accepting this paper. Therefore, I recommend rejection.

**Reviewer Concerns:**

Regarding addressed concerns, the authors effectively responded to queries about the multiple method versions in Section 3.4, clarifying that diverse variants are a strength allowing practitioners to balance compute constraints and performance, with expressive methods outperforming simpler ones. They also resolved doubts about computational complexity, explaining that mixture-based merging reduces the exponential cost of retraining to polynomial, and checkpointing can cut the cost of $K$ components to near $O(T)$. Additionally, the authors added a new Section 3.4 to theoretically link posterior flexibility and Pareto front accuracy, and confirmed the framework can handle Pareto front constraints.

As for outstanding concerns, reviewers’ doubts about Hessian quality remain unaddressed, as no controlled study was provided to link Hessian approximation accuracy with Pareto front error. The issue of off-manifold solutions in scenarios like strong parameter non-identifiability in LLMs is also unresolved, since the authors only acknowledged it without delineating failure modes or detection heuristics. Moreover, the authors did not supplement apples-to-apples wall-clock time comparisons for upfront training costs, and one reviewer still contested that AdamW’s second-moment information does not equate to Hessian approximation.

Moreover, this paper has not compared with multiojective baselines especially for those aiming to recover the Pareto front. Only comparing multitask finetuning is not convincing for the advantage of computational cost in the total workflow.

**Reviewer Scores:**

No

---

### Decision · Program_Chairs · 2026-01-26

Reject